# InfoGround: Ground Manipulation Concepts with Maximal Information Boost

## Abstract

We aim at *grounding* manipulation concepts proposed by Large Language Models in the form of task-related step-by-step instructions to their corresponding physical states, i.e., *key states*, from *unannotated* demonstrations. The grounded concepts facilitate training efficient manipulation policies and promote generalization. Current methods mainly rely on multimodal foundation models to localize these key states, which involves encoding physical observations and textual descriptions and measuring their feature similarity. *However,* the grounding often lacks *accuracy* and *semantic consistency* due to the limited availability of curated data for multimodal encoders and variations in physical states. To effectively leverage the commonsense knowledge embedded within pre-trained foundation models, we introduce an information-theoretic criterion designed to enhance grounding efficiency *without* requiring costly fine-tuning. Our approach is based on the observation that the uncertainty of a state *diminishes rapidly* as it approaches a key state, since this state admits more physical constraints than non-key states. This phenomenon is characterized as *maximizing* the increase rate in mutual information between the key state and its preceding states, referred to as *Maximal Information Boost (MIB)*. By employing MIB, we can train a key state grounding network to effectively utilize noisy similarity measures from existing multimodal encoders. Experiments demonstrate that our grounded key states exhibit superior semantic compatibility with the instructions. *Furthermore,* when used as sub-goal guidance, our grounding method generates key states leading to manipulation policies with higher success rates and better generalization across various robotic tasks, verifying the significance of the grounded concepts.

## 1 Introduction

An embodied agent should be capable of perceiving its environment, performing reasoning, and taking actions to interact with the physical world and human agents. The success of deep learning has significantly contributed to the development of such intelligent embodied agents. For example, we have neural networks for detection, recognition, and segmentation in computer vision, which even demonstrate human-level perception in specific benchmarks. Additionally, considerable effort has been made in training action policies using techniques like deep reinforcement learning, yielding notable progress. Nonetheless, providing embodied agents with human-level reasoning abilities was challenging until the recent advancements in Large Language Models (LLMs).

Leveraging predictive training of sophisticated network architectures with internet-scale text corpora, LLMs have shown various emergent capabilities, e.g., commonsense reasoning and sub-goal-based long-horizon planning, critical to enabling an embodied agent for intelligent interaction. For example, given a task description or prompt, an LLM could speed out a set of step-by-step instructions (sub-goals or concepts) that, when executed successfully, accomplish the desired task. A major *obstacle* to fully utilizing LLMs for embodied tasks is the lack of *grounding* in LLM-generated instructions. This grounding instantiates related *low-level action policies* and determines whether the *goal state* has been achieved. One possible approach is to select a finite set of manipulation concepts, train corresponding policies, and then constrain the LLMs' planning to these limited concepts, which may commit to sub-optimality. Another is to train vision-language models (VLMs) to assess the compatibility of the current state with the generated concepts, which achieves grounding more flexibly by eliminating the dependence on a limited concept vocabulary. *However,* the grounding of

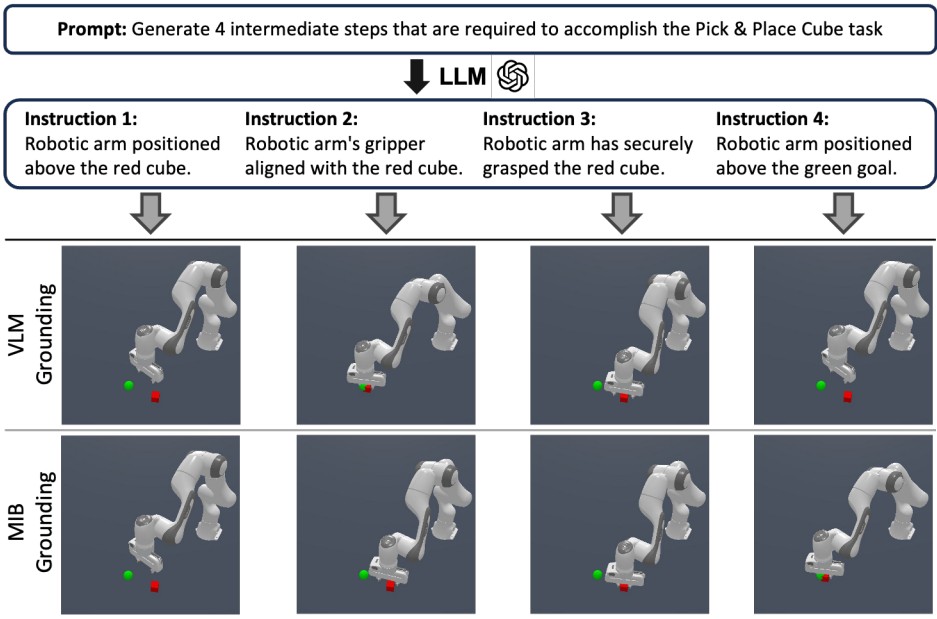

Figure 1: LLM-generated manipulation concepts or instructions and their grounding using multi-modal encoders (VLM Grounding), which shows noisy localization (third column) and misalignment in semantics (last column). The grounding achieved with the proposed Maximal Information Boost criterion demonstrates better correspondence (MIB Grounding), which is critical for learning effective manipulation policies.

current VLMs still falls short due to *ambiguities* in text descriptions in their training data and the *limited granularity* required for accurate correspondence, as evidenced in Fig. 1.

We aim to leverage the flexibility of vision-language models (VLMs) for concept grounding while focusing on *enhancing* their *robustness* and *accuracy*. We propose a *global* optimization framework instead of collecting more multimodal training data for VLMs and resorting to fine-tuning that subjects to catastrophic forgetting. This framework trains a grounding network to select grounded physical states from noisy demonstrations. To achieve this, we develop an *information-theoretic metric* that encourages grounding to states with *sufficient* physical significance. In other words, the grounded states should admit more physical constraints (otherwise, they could be a random point in the air), which translates into a rapid increase in the mutual information between the grounded state and its preceding states. We call this metric *Maximal Information Boost (MIB)* and combine it with the noisy compatibility scores provided by VLMs to formulate a training objective for the grounding network. After training, the grounding network takes in observations from a demonstration and an LLM-generated manipulation concept, outputting the grounded state that optimizes both VLM compatibility and the proposed MIB metric.

Our experiments demonstrate that the grounding network trained using the Maximal Information Boost can predict physical states that *align* well with the generated query concepts, eliminating the need for human supervision or heuristic rules. *Furthermore,* we explicitly test its efficacy using grounded manipulation concepts to guide the manipulation policy training. We show that the grounded concepts effectively *mitigate* compounding errors in manipulation tasks, and the resulting policies significantly *outperform* state-of-the-art algorithms in complex robotic manipulation tasks with nice *generalization*. These results confirm the potential of the proposed Maximal Information Boost in enhancing VLM-based manipulation concept grounding and endowing embodied agents with robust and generalizable manipulation capabilities.

## 2 METHOD

We now detail the problem and introduce the employed compatibility function, followed by the proposed Maximal Information Boost criterion that leads to the grounding objectives.

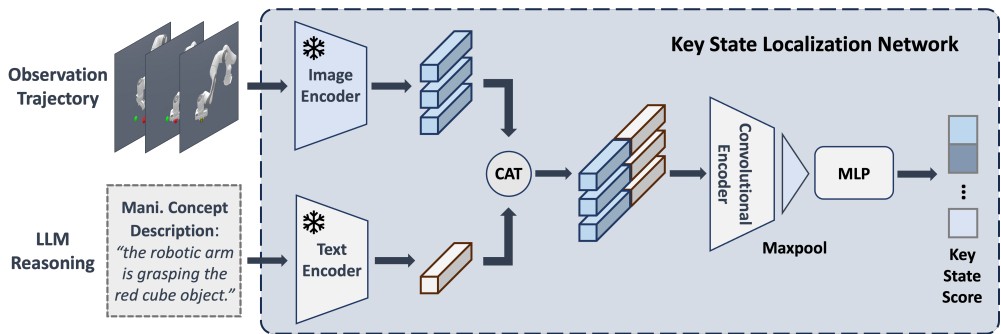

Figure 2: The proposed Key State Localization Network (KSL-Net) for manipulation concept grounding. Images from a demonstration ($\tau^i$) are mapped to state vectors ($s_t^i$) by an image encoder, and the key state (concept) description ($\mu_k$) is converted to an embedding by a text encoder. Both encoders are fixed. The text embedding ($e_k$) is then appended to all state vectors. These augmented state vectors are further processed by a fully convolutional encoder and a multi-layer perceptron (MLP) to derive the probability ($\mathrm{p}_k^i$) of each state being the grounded key state.

**Problem Statement**  We aim at localizing a set of physical states corresponding to a sequence of textual descriptions of the manipulation concepts, i.e., *key states*, which a trajectory should hit in order to accomplish a task. More specifically, given a dataset of $N$ demonstrations $\mathcal{D} = \{\tau^i\}_{i=1}^N$ for a task $\mathcal{T}$, where $\tau^i = \{(s_t^i, a_t^i)\}_{t=1}^{T_i}$, we will *first* leverage an LLM (OpenAI, 2023) to determine the set of key states sequentially involved in performing the task and their corresponding textual descriptions, i.e., $\mathcal{T} = \{\mu_k\}_{k=1}^K$. Our goal is to find the $N \times K$ temporal indices $\{t_k^i\}_{i,k=1}^{N,K}$, such that $s_{t_k^i}^i$ is the (key) state in trajectory $\tau^i$ that corresponds to the $k$-th textual description $\mu_k$. An intuitive way of computing an index is to map $\mu_k$ and the physical states $\{s_t^i\}$ to a shared feature space via multimodal encoders, e.g., CLIP (Radford et al., 2021), and then select the state that maximizes the similarity. *However*, as shown in Fig. 1, the individually determined key states are often inaccurate and lack consistency across different trajectories. Instead of independently checking the similarity scores, we propose a global optimization framework that trains a neural network to localize these key states with the following criteria:

- *Semantic Compatibility*: The semantic meaning of the selected key state should be compatible with the textual description.

- *Chronological Correctness*: The selected key states, when arranged in the order of the textual descriptions, shall respect their order in the trajectory.

- *Maximal Information Boost*: The mutual information between the key state and its precedents should *maximally increase* when compared to other non-key states.

The first models the likelihood of a state being semantically meaningful, while the last provides an information-theoretic prior on a state being physically critical. Next, we specify the key state localization network and elaborate on each criterion to derive the training objective that grounds the LLM-generated concept descriptions.

## 2.1 Key State Localization Network for Grounding

We denote the localization network as $\phi$, which takes as input a trajectory $\tau^i$ and the embedding $e_k = \psi(\mu_k) \in \mathbb{R}^P$ of a key state description. We abuse notation and denote $s_t^i \in \mathbb{R}^Q$ as both the physical state and its embedding. After appending the key state embedding $e_k$ to every state $s_t^i$ in the trajectory, the neural network $\phi$ maps the augmented physical states into a distribution corresponding to the key state selection probability, i.e., $\phi : \mathbb{R}^{(Q+P) \times T} \to \mathbb{R}^T$ and $T$ is the length of the trajectory. We can also write $\mathrm{p}_k^i = \phi(\tau^i, \mu_k) \in \mathbb{R}^T$ for the probability of each state in $\tau^i$ being the key state described by concept $\mu_k$. In the ideal case, $\mathrm{p}_k^i$ should be an indicator function; thus, we will penalize its entropy during training. The overall structure of the proposed Key State Localization Network (KSL-Net) is shown in Fig. 2. A convolutional encoder is used to fuse the information from different modalities, followed by a max-pooling to aggregate information from different states and an MLP to output the key state probability.

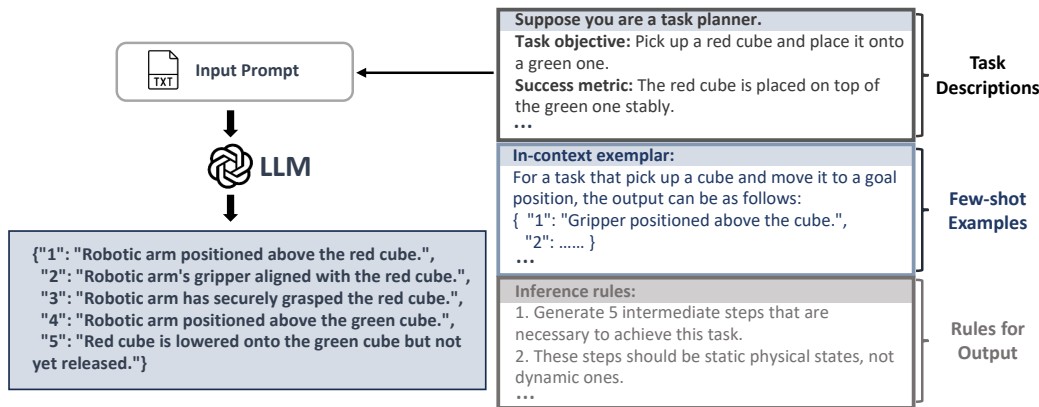

Figure 3: Prompting a large language model to generate task-related step-by-step instructions (manipulation concepts) that focus on describing the physical states critical for accomplishing the task.

## 2.2 SEMANTIC COMPATIBILITY WITH PARAPHRASING

In the following, we describe how to apply pre-trained multimodal foundation models to construct the compatibility functions between key-state descriptions and physical states. Given a task $\mathcal{T}$ in text, we first ask an LLM to produce a set of sequential instructions, which, if executed, can complete the task (Ouyang et al., 2022; Wei et al., 2021; Yu et al., 2023). Since our target is the set of states whose completion is critical to the success of the task, we explicitly enforce the LLM to give instructions that focus on describing specific physical states. For example, prompting LLMs without emphasis on physical states often results in general and non-specific instructions, such as "Plan a path to move the cube to the green target position." In contrast, a well-designed prompt can guide LLMs to generate instructions more relevant to describing a key physical state, such as "Robotic arm has securely grasped the red cube." The above prompt engineering is detailed in Fig. 3, where we employ the self-consistency chain-of-thoughts technique (Wang et al., 2022) to achieve more "physical" descriptions of the key states.

After we get the set of key state descriptions $\mathcal{T} = \{\mu_k\}_{k=1}^{K}$, we can employ a pair of encoders $\psi^s$ and $\psi^\mu$ (Radford et al., 2021) to map a state and a description to a shared latent space, which is inherently rich in semantic information, thereby reducing the gap between the visual and textual domains. A basic version of the compatibility function would be:

$$f(s_t, \mu_k) = \psi^s(s_t) \cdot \psi^\mu(\mu_k). \tag{1}$$

Here we consider a single trajectory $\tau$ and omit the trajectory index. As illustrated in Fig. 1, if we simply select the corresponding key state for $\mu_k$ by maximizing $f$ over $t$, we can get very noisy estimates. The underlying reason can be three-fold: 1) there are multiple ways to describe an image (state), and the training data for $\psi^{s,\mu}$ may not be inclusive; 2) there may be a domain gap between the training images for the encoders and the images from the trajectory, and 3) the text encoder may be trained on not so physically-relevant descriptions. To alleviate this, we propose to perturb the input to the text encoder without changing its intrinsic information and then apply an average over different perturbed versions to help smooth out the noise.

By leveraging various prompts for the LLM, one can obtain diverse yet semantically consistent paraphrases for $\mu_k$. For example, given the instruction "Robotic arm has securely grasped the red cube," alternatives like "The red cube is securely in the grip of the robotic arm" can be available. Please check Sec. A.9 for more details. With the perturbations, i.e., $\mu_k \to \{\mu_{k,m}\}_{m=1}^{M}$, we can write the enhanced compatibility function as following:

$$f^c(s_t, \mu_k) = \frac{1}{M} \sum_{m=1}^{M} \text{Softmax}\langle f(s_1, \mu_{k,m}), f(s_2, \mu_{k,m}), ..., f(s_T, \mu_{k,m})\rangle[t], \tag{2}$$

which averages the compatibility over different variants of the key state description, mitigating alignment errors between the two modalities and resulting in a smoother compatibility function.

## 2.3 MAXIMAL INFORMATION BOOST AT KEY STATE

Even though the enhanced compatibility function $f^c$ exhibits better smoothness, the localization accuracy still lags, given the limited annotation granularity (e.g., insufficient fine-grained robot motion labels). On the other hand, key (physical) states exist on their own without attaching to human semantics. We propose that the reason we assign a certain physical state a "name" resides in the fact that the state itself possesses physical significance. More explicitly, according to our observations, we find that the uncertainty of the state

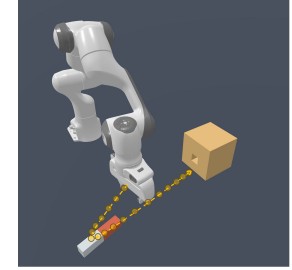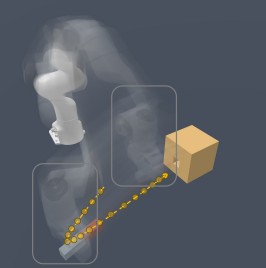

Figure 4: Left: trajectory of the end-effector. Right: the average of the images over the entire horizon.

quickly diminishes when getting close to the moment a key state is achieved. For example, in Fig. 4 (right), the average image of the demonstration is much more blurred (uncertain) in the region distant to the key state (e.g., "block is firmly grasped"). It also shows that the uncertainty gets much smaller near the key state, i.e., the overlap image is sharper. Please see Sec. A.2 for a more detailed illustration. Next, we characterize these observations using information-theoretic metrics.

**Information Boost** We denote a key state random variable (RV) as $s_{t_k}$, which represents the physical state observed at the moment when the key state description $\mu_k$ is true and can be sampled by instantiating a trajectory $\tau^i$. Similarly, we write its precedent state variable as $s_{t_k-\Delta t}$ with $\Delta t > 0$. Then the Shannon's mutual information $\mathbb{I}(s_{t_k}; s_{t_k-\Delta t})$ measures how much we know (uncertainty reduction) about $s_{t_k-\Delta t}$ after knowing $s_{t_k}$. Our observations for key states are:

a. The quantity $\mathbb{I}(s_t; s_{t-\Delta t})$ increases when approaching a key state, i.e., $t$ getting close to the key moment $t_k$. This is generally true since a key state admits more physical constraints than a non-key state and confines the dynamics nearby.

b. Moreover, $\mathbb{I}(s_t; s_{t-\Delta t})$ is maximized at the key moment $t = t_k$, since the constraints that guide the actions are satisfied and become ineffective when the key state is achieved. In other words, $s_{t_k}$ confines more the state $s_{t_k-\Delta t}$ than $s_{t_k+\Delta t}$.

c. Besides being maximized, $\mathbb{I}(s_t; s_{t-\Delta t})$ should also be significant at $t_k$ in the sense that a distant precedent state $s_{t_k-\Delta t'}$ ($\Delta t' > \Delta t$) shall be loosely informed by $s_{t_k}$. This helps alleviate ambiguity in the localization. For example, if a gripper stays steady after grasping the mug handle (before heading to the faucet), then any state during this period could maximize $\mathbb{I}(s_t; s_{t-\Delta t})$; however, only the state at the very moment of "grasping" shall be treated as significant.

Given the above observations, we formalize a key state as one that maximizes the following quantity:

$$\mathcal{L}^{\mathrm{IB}}(s_t) = \mathbb{I}(s_t; s_{t-\Delta t}) - \mathbb{I}(s_t; s_{t-\Delta t'}), \tag{3}$$

where $\Delta t$ and $\Delta t'$ are operational parameters whose role will be studied in the experiments. We name the quantity in Eq. 3 as the *Information Boost* of a state $s_t$, as it measures the boost in the mutual information between the anchor state ($s_t$) and its precedents, i.e., from $s_{t-\Delta t'}$ to $s_{t-\Delta t}$.

By maximizing the information boost, we can therefore locate the state $s_t$ that satisfies the aforementioned observations, which are meant to be the characteristics of a key state. Thus, we propose that a key state should come with the *Maximal Information Boost*. Moreover, to make Eq. 3 computable, we write it using the Shannon entropy as follows:

$$\mathcal{L}^{\mathrm{IB}}(s_t) = \mathbb{H}(s_{t-\Delta t}) - \mathbb{H}(s_{t-\Delta t}|s_t) - (\mathbb{H}(s_{t-\Delta t'}) - \mathbb{H}(s_{t-\Delta t'}|s_t)). \tag{4}$$

With the assumption that the unconditional RV $s_t$ is locally stationary, as evidenced in Sec. A.1, we have:

$$\mathcal{L}^{\mathrm{IB}}(s_t) = \mathbb{H}(s_{t-\Delta t'}|s_t) - \mathbb{H}(s_{t-\Delta t}|s_t) \tag{5}$$

Next, we elaborate on the training objectives for key state grounding derived from the enhanced compatibility function and the maximal information boost criterion.

## 2.4 KEY STATE GROUNDING

As discussed, the selected key states $\{t_k^i\}_{i,k=1}^{N,K}$ from $\mathrm{p}_k^i$ should maximize the enhanced compatibility with the descriptions $\mu_k$ as well as the information boost that measures their physical criticality. To simplify notation, we assume that all trajectories have the same length $T$ (after normalization, details can be found in Sec. A.4) and denote $\boldsymbol{T} = \{1, 2, ..., T-1, T\}$ as the temporal index for each state. The estimated index for trajectory $\tau^i$ of key state $\mu_k$ is then $\hat{t}_k^i = \boldsymbol{T}\mathrm{p}_k^i$. With these, we can write the compatibility loss as:

$$\mathcal{L}^{\mathrm{comp}}(\phi; \mathcal{D}) = -\sum_{i,k=1}^{N,K} f^{\mathrm{c}}(s_{\hat{t}_k^i}^i, \mu_k). \tag{6}$$

We further approximate the entropy in Eq. 3 with the variance of the states and derive the following information boost loss:

$$\mathcal{L}^{\mathrm{infoboost}}(\phi; \mathcal{D}) = \sum_{k=1}^{K} [\mathrm{Var}(\{s_{\hat{t}_k^i - \Delta t}^i\}_{i=1}^N) - \mathrm{Var}(\{s_{\hat{t}_k^i - \Delta t'}^i\}_{i=1}^N)] \tag{7}$$

Additionally, we have two other terms that encourage $\mathrm{p}_k^i$ to be as concentrated as possible and $\hat{t}_k^i$ to increase as $k$ increase (i.e., the chronological correctness of the estimated indices):

$$\mathcal{L}^{\mathrm{ent}}(\phi; \mathcal{D}) = \sum_{i,k=1}^{N,K} \mathbb{H}(\mathrm{p}_k^i); \quad \mathcal{L}^{\mathrm{order}}(\phi; \mathcal{D}) = \max((\hat{t}_k^i - \hat{t}_{k+1}^i + n), 0). \tag{8}$$

Here $n < 5$ is a number that helps enforce a small margin between two consecutive key states. The **final training loss** is:

$$\mathcal{L} = \mathcal{L}^{\mathrm{comp}}(\phi; \mathcal{D}) + \lambda_i \mathcal{L}^{\mathrm{infoboost}}(\phi; \mathcal{D}) + \lambda_e \mathcal{L}^{\mathrm{ent}}(\phi; \mathcal{D}) + \lambda_o \mathcal{L}^{\mathrm{order}}(\phi; \mathcal{D}), \tag{9}$$

with $\lambda$'s the weighting between different terms and are studied in the experiments. We name the proposed method *InfoGround* as its resulting groundings characterize the Maximal Information Boost.

## 2.5 CONTROL POLICY TRAINING

After key state grounding, each demonstration in the training set has its corresponding key states. To incorporate key state information for task completion, we leverage the CoTPC framework (Jia et al., 2023) to concurrently perform action prediction and the dynamic estimation of key states during the training phase. The overall objective function of policy network training is as follows,

$$\mathcal{L}_{\mathrm{total}} = \underbrace{\mathbb{E}_{\tau \in D} \mathcal{L}_{\mathrm{bc}}(\hat{a}_t, a_t)}_{\text{behavior cloning loss}} + \underbrace{\frac{\lambda_k}{K} \sum_{k=0}^{K-1} \mathbb{E}_{\tau \in D} \mathcal{L}_{\mathrm{ks}}(\hat{s}_k^i, s_k^i)}_{\text{key state prediction loss}}$$

An additional key state prediction loss is appended on the basis of the traditional behavior cloning loss, which is used to implicitly guide the policy network step by step to reach the key states.

## 3 EXPERIMENTS

In this section, we thoroughly evaluate the proposed key state grounding strategy, InfoGround. We first compare its effectiveness against baselines in four complex multi-step tasks from ManiSkill2 (Gu et al., 2023), and then assess its performance and generalization in novel scenarios with unseen configurations and objects. Additionally, an ablation study explores various key state grounding approaches. Further details of the control policy training can be found in Sec. A.6.

### 3.1 EXPERIMENTAL SETUP

**Baselines** The evaluation and comparison are performed with the following major baselines:

- Behavior Transformer (BeT) (Shafiullah et al., 2022): Behavior Transformer utilizes a modified transformer structure with action discretization and multi-task action correction to model multimodal continuous action sequences from unlabeled demonstrations.

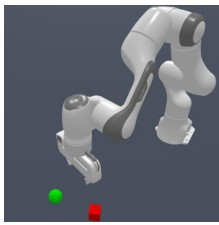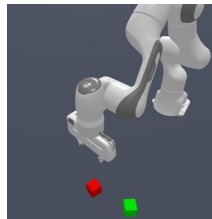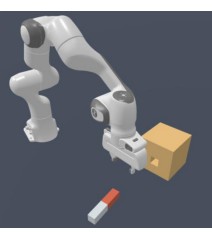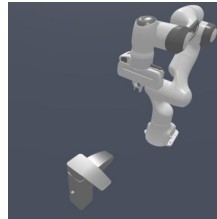

Figure 5: The four complex, multi-step object manipulation tasks involved in our experiments, as shown from left to right: Pick and Place Cube, Stack Cube, Peg Insertion Side, and Turn Faucet.

- Decision Diffuser (DD) (Ajay et al., 2023): The Decision Diffuser, inspired by the classifier-free diffusion model (Ho & Salimans, 2022), offers a refined generative approach for sequential decision-making, promoting novel behaviors.
- Decision Transformer (DT) (Chen et al., 2021): Decision Transformer leverages a causally masked Transformer to solve offline reinforcement learning problems by autoregressively modeling the densely-labeled rewards and optimal actions.
- Chain-of-Thought Predictive Control (CoTPC) (Jia et al., 2023): Chain-of-Thought Predictive Control utilizes a set of manually labeled key states to identify short subsequences, i.e., "chains-of-thought," that mark the completion of subgoals in complex tasks.

**Multi-Step Manipulation Tasks**  We adopt the experimental setting used in the most recent state-of-the-art (Jia et al., 2023) and choose four complex multi-step object manipulation tasks from Maniskill2 environments (Gu et al., 2023). These four tasks are shown in Fig. 5:

**Experimental Setting**  Our policy training follows the behavioral cloning (BC) paradigm without densely labeled rewards. Similar to CoTPC, we set the state space as the observation space, assuming full observability unless stated otherwise. For fair comparisons, we utilize the same set of 500 randomly sampled successful trajectories for a given task. These trajectories, varying in initial states and environmental configurations, are sourced from multiple experts, adding diversity and randomness. This variability makes them an effective benchmark for assessing the InfoGround algorithm's capability in robust and accurate key state localization. Next, we detail the evaluation metric and results.

## 3.2 RESULTS

**Evaluation Metric**  Our primary evaluation metric is the task success rate. The task completion criteria are detailed in Sec. 3.1 above. For complex multi-step tasks, we also report the success rate of completing intermediate sub-tasks. For instance, consider the Peg Insertion Side task. The ultimate objective is to insert the peg into a horizontal hole in the box. Intermediate metrics for this task include successful peg grasping by the robotic arm and alignment of the peg with the hole. During the evaluation phase, both training-exposed and novel environmental configurations, such as different initial joint positions of the robot, are tested. For the Turn Faucet task, we conducted additional tests on faucets with geometric structures not present in the training set.

**Main Results**  Tab. 1 and Tab. 2 show our results for the seen and unseen environmental configurations (including zero-shot with novel objects), respectively. Tab. 1 demonstrates the superior performance of our proposed method in comparison to various baseline methods across multiple tasks. Notably, while some baselines struggle with complex multi-step tasks, our method consistently outperforms them, verifying the effectiveness of the proposed InfoGround for localizing physically meaningful key states. Due to the poor performance of various conventional methods in dealing with novel configurations, we report the results for the two most effective baselines for comparative evaluation in unseen scenes. As evidenced in Tab. 2, our approach still outperforms these baselines, suggesting that the key states grounded using InfoGround significantly enhance the generalization of the trained manipulation policies in new scenarios.

## 3.3 ABLATION STUDY

**Different Key State Selection**  We conducted ablation studies on key state selection strategies to validate our proposed key state grounding. The results are presented in Tab. 3. The strategies

Table 1: Comparison of Success Rates (SR) between our approach and various baselines across multiple tasks, evaluated on initial environmental configurations seen during training. For more complex tasks like Peg Insertion, success rates for intermediate sub-tasks such as "grasp" and "align" are also reported.

| METHODS | P&P CUBE | STACK CUBE | TURN FAUCET | PEG INSERTION | | | MEAN (%) |
|---|---|---|---|---|---|---|---|
| | TASK SR | TASK SR | TASK SR | GRASP SR | ALIGN SR | INSERT SR | TASK SR |
| BeT | 23.6 | 1.6 | 16.0 | 90.0 | 17.0 | 0.8 | 10.5 |
| DD | 11.8 | 0.6 | 53.6 | 86.8 | 9.2 | 0.6 | 16.7 |
| DT | 65.4 | 13.0 | 39.4 | 97.8 | 41.8 | 5.6 | 30.9 |
| MASKDP+GT GOALS | 54.7 | 7.8 | 28.8 | 62.6 | 5.8 | 0.0 | 22.8 |
| CoTPC | 75.2 | 58.8 | 56.4 | 99.6 | 98.2 | 52.8 | 60.8 |
| OURS | **89.4** | **83.2** | **67.6** | **100.0** | **100.0** | **75.4** | **78.9** |

Table 2: Comparison of Success Rates (SR) between our approach and various baselines across multiple tasks, evaluated on environmental configurations unseen during training. For more complex tasks like Peg Insertion, success rates for intermediate sub-tasks such as "grasp" and "align" are also reported. For the Turn Faucet task, we evaluated unseen environmental configurations and faucets with novel geometric structures (zero-shot).

| METHODS | P&P CUBE (UNSEEN) | STACK CUBE (UNSEEN) | TURN FAUCET (UNSEEN & 0-SHOT) | | PEG INSERTION (0-SHOT) | | |
|---|---|---|---|---|---|---|---|
| | TASK SR | TASK SR | TASK SR | Task SR | Grasp SR | Align SR | Insert SR |
| DT | 50.0 | 7.0 | 32.0 | 9.0 | 92.3 | 21.8 | 2.0 |
| CoTPC | 70.0 | 46.0 | 57.0 | 31.0 | 95.3 | 72.3 | 16.8 |
| OURS | **79.0** | **64.0** | **61.0** | **42.0** | **98.3** | **81.8** | **21.5** |

"Last State" and "All States" denote selecting only the final state and all states as key states. These approaches exhibit poor performance as they fail to utilize the hierarchical information typical of complex multi-step tasks. "Manual Rules" refers to a set of predefined key state selection rules, consistent with the settings in (Jia et al., 2023). These results verify the importance of proper key state selection and confirm the usefulness of the key states from InfoGround.

**Effectiveness of MIB** We verify the effectiveness of the proposed MIB criterion in improving the VLM-grounding quality by training InfoGround with four different VLMs: CLIP (Radford et al., 2021), BLIP (Li et al., 2023), BLIP2 (Li et al., 2023), and FLAVA (Singh et al., 2022). The evaluations are reported in Tab. 4. The label "Only" refers to key state selections based solely on the VLM-compatibility scores, without further applying Maximal Information Boost as described in Sec. 2.3. It is unanimously observed that by enforcing MIB in the grounding process, we can obtain better grounded key states measured by the success rate of the learned manipulation policies, demonstrating the effectiveness of MIB.

Table 3: Success rates of various key state selection strategies evaluated on initial environmental configurations seen during training. Additionally, we report success rates of intermediate sub-tasks for the Peg Insertion task.

| SELECTION | STACK CUBE | PEG INSERTION | | |
|---|---|---|---|---|
| | TASK SR | Grasp SR | Align SR | Insert SR |
| LAST STATE | 12.0 | 96.8 | 65.6 | 9.6 |
| ALL STATES | 30.0 | 92.4 | 70.2 | 18.0 |
| MANUAL RULES | 58.8 | 99.6 | 98.2 | 52.8 |
| OURS | **83.2** | **100.0** | **100.0** | **75.4** |

## 4 RELATED WORK

**Large-scale vision-language model.** Large Language Models (LLMs) possess extensive internal knowledge about the world due to their large-scale pertaining (Brown et al., 2020; Touvron et al., 2023). Concurrently, Foundation models have excelled in various computer vision downstream tasks. (He et al., 2022; Dosovitskiy et al., 2020; Oquab et al., 2023). These advancements have

Table 4: Success rates for different VLM-Groundings with and without Maximal Information Boost.

| Methods | P&P CUBE | | STACK CUBE | | TURN FAUCET | | PEG INSERTION | |
|---|---|---|---|---|---|---|---|---|
| | Seen | Unseen | Seen | Unseen | Seen | Unseen | Seen | Unseen |
| OnlyClip | 73.2 | 65.0 | 66.0 | 51.0 | 56.0 | 40.0 | 57.0 | 7.75 |
| Clip+MIB | 78.8 | 71.0 | **76.4** | **56.0** | 61.0 | **60.0** | **75.0** | **21.5** |
| OnlyBlip | 69.0 | 64.0 | 52.8 | 26.0 | 53.8 | 44.0 | 52.6 | 11.3 |
| Blip+MIB | 85.0 | **73.0** | 68.8 | 50.0 | 61.6 | 59.0 | 71.6 | 20.0 |
| OnlyBlip2 | 73.0 | 59.0 | 56.6 | 43.0 | 54.6 | 51.0 | 59.4 | 11.3 |
| Blip2+MIB | 76.8 | 68.0 | 70.0 | 52.0 | 56.4 | 57.0 | 70.8 | 20.8 |
| OnlyFlava | 74.0 | 59.0 | 40.2 | 28.0 | 55.0 | 56.0 | 63.6 | 13.0 |
| Flava+MIB | **83.0** | **73.0** | 72.0 | 54.0 | **65.0** | **60.0** | 69.0 | 18.3 |

spurred robust cross-modal architectures, with Vision-Language Models (VLMs) excelling in various tasks, including visual question answering (Wang et al., 2021; Zeng et al., 2022), image captioning (Yang et al., 2022; Li et al., 2023), and image-to-text retrieval (Singh et al., 2022; Wang et al., 2023a). Thesse VLMs bridge the visual-textual gap (Radford et al., 2021; Li et al., 2022), enhancing cross-modal applications (Rombach et al., 2022; Zhang & Agrawala, 2023). Cross-modal proficiency allows for natural language understanding and visual expression identification, promising practical applications for the internalized knowledge of these models.

**Learning from Demonstrations.** The goal of Learning from Demonstration (LfD) is to teach agents complex tasks through expert demonstrations, avoiding costly self-exploration (Schaal, 1996; Ravichandar et al., 2020). LfD methods include Inverse Reinforcement Learning (IRL), online Reinforcement Learning (RL) with demonstrations, and Behavior Cloning (BC) (Argall et al., 2009). IRL infers reward functions from observed behaviors, though it can be computationally demanding. (Wulfmeier et al., 2015; Finn et al., 2016; Zakka et al., 2022). Online RL with demonstrations combines dynamic online RL with offline guidance from demonstrations. (Ho & Ermon, 2016; Riedmiller et al., 2018; Stooke et al., 2021). BC employs supervised learning to map input states to actions but may face problems like sparse examples and imitation errors. (Laskey et al., 2017; Sasaki & Yamashina, 2020; Zeng et al., 2021; Shridhar et al., 2023). Our proposed approach aims to minimize imitation errors by focusing on key steps for task accomplishment.

**Foundation models for task planning.** Some studies investigate the use of foundation models (Yang et al., 2023), such as Large Language Models (LLMs), for general task planning, excelling in code generation and leveraging open-source tools for diverse tasks. (Gupta & Kembhavi, 2023; Surís et al., 2023; Liu et al., 2023b; Shen et al., 2023). Moreover, LLMs enable upstream compositional planning for creative content generation (Liu et al., 2023a; Feng et al., 2023). In embodied planning, a challenge is enabling agents to interact with diverse real-world environments (Bommasani et al., 2021; Fang et al., 2020). Foundation models drive research in converting language instructions and environmental cues into control signals (Huang et al., 2023a; Driess et al., 2023). Additionally, LLMs and VLMs identify reward or value functions using carefully-crafted prompting strategies (Yu et al., 2023; Huang et al., 2023b; Wang et al., 2023b). Our work, unlike these studies, leverages foundation models to plan essential steps for successful task completion.

## 5 DISCUSSION

We propose a novel method to ground manipulation concepts with key states from unannotated demonstrations, leveraging LLMs for high-level task planning and pre-trained multimodal encoders for semantic compatibility. By introducing the maximal information boost (MIB), we ensure these grounded states are physically significant, reducing noise from the compatibility measures. Our method performs effectively on four challenging manipulation tasks, indicating its ability to reduce compounding errors in complex robotic manipulations. The efficacy of the MIB criterion is also verified with different VLMs, demonstrating its generalizability. We expect the proposed to endow embodied agents with the capability to learn more accurate concept grounding for intelligent interactions in future research.

**Ethics Statement**    In our work, we primarily focus on using foundation models and information-theoretic metrics to ground manipulation concepts. Prior research has raised concerns about potential biases and stereotypes in some foundation models. The community has since witnessed an increasing emphasis on aligning foundation models with human values and ensuring safety. It's noteworthy that our research specifically focuses on the embodied agent domain, i.e., aligning human concepts with robot manipulation concepts. Thus, we believe that enhancing the accuracy of concept grounding does reduce ethical concerns.

**Reproducibility Statement**    Experiments included in this paper are conducted over the same random seed for all tasks, thus the reproducibility and reliability are guaranteed. The hyperparameters and optimization details of KSL-Net are included in Sec. A.5. The hyperparameters and implementation details of policy training are also included in Sec. A.6. Our code and trained models will be released after the review process.

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

# A APPENDIX

## A.1 LOCAL STATIONARITY OF $s_t$

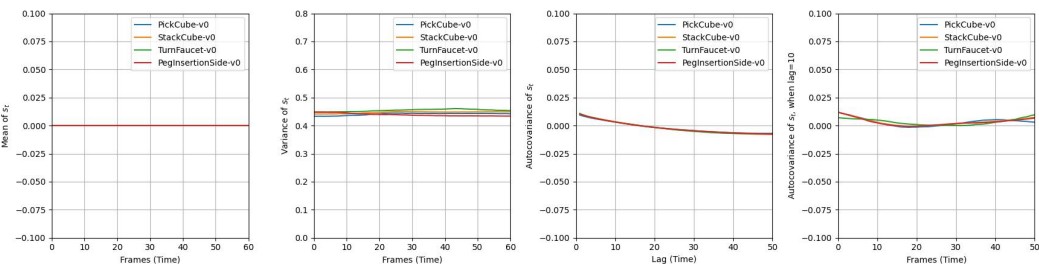

Figure 6: For all four tasks, the mean stationarity, variance stationarity, and autocovariance stationarity of $s_t$ are considered.

We consider the state $s_t$ as an unconditional random variable. Empirical evidence supporting the local stationarity of the random variable $s_t$ is depicted in Fig. 6. We provide an interpretation of these figures in a sequential manner from left to right.

- The first figure demonstrates the mean stationarity of $s_t$: the time series exhibits a consistent mean across all time points, denoted as $E[\xi_t] = \mu$, with $\mu$ being a time-invariant constant. Notably, $\mu$ here approximates 0.

- The second figure underscores the variance stationarity of $s_t$, expressed as $\text{Var}[\xi_t] = \sigma^2$, In this context, $\sigma^2$ is a constant not dependent on $t$. The variance here approximates 0.45.

- The third figure demonstrates the autocovariance stationarity of $s_t$. Given a specific lag $h$, the autocovariance depends solely on $h$, decoupled from the temporal variable $t$. This implies that $\text{Cov}(\xi_t, \xi_{t+h})$ operates solely as a function of $h$ without any independence on $t$. As the lag $h$ varies, the autocovariance $\text{Cov}(\xi_t, \xi_{t+h})$ consistently approximates 0.

- The fourth figure complements the autocovariance stationarity of $s_t$. We choose a case where lag = 10, demonstrating that the autocovariance $\text{Cov}(\xi_t, \xi_{t+h})$ is almost independent of time $t$, remaining consistently close to 0.

The mean stationarity, variance stationarity, and autocovariance stationarity of $s_t$ ensure that this trajectory time series approximates local stationarity.

## A.2 MAXIMAL INFORMATION BOOST ILLUSTRATION

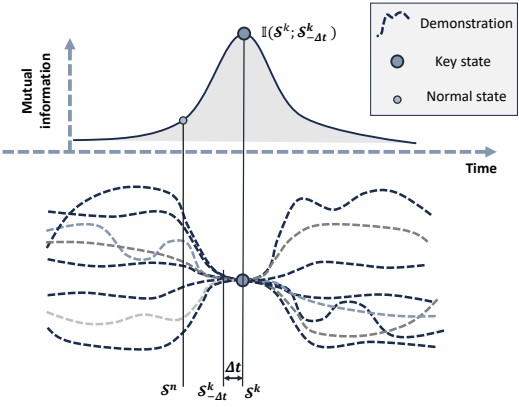

Figure 7: Illustration of the mutual information peaking near the key state

Fig. 7 illustrates that for the behavior cloning task, the pre-collected demonstrations are perturbed and noisy. However, intricate low-level manipulation tasks can typically be decomposed into sub-steps using key states. These key states are vital for task completion; in other words, failing to achieve a key state makes it challenging for the agent to proceed, leading to the accumulation of compounding errors.

For instance, for tasks such as *PickCube* and *StackCube*, the initial step involves the robotic arm grasping the cube, a prerequisite for further actions. Thus, states semantically representing "the robotic arm grasping the cube" can be defined as key states. Within the semantic latent space, the semantic embedding vectors of key states $s^k$ tend to occupy a relatively stable position. On the other hand, $s^k_{-\delta t}$ representing a state slightly before $s^k$, is restricted from significant shifts. This inherent constraint ensures the feasibility of trajectories effectively reaching key states with critical physical semantics. Consequently, we claim that the mutual information $\mathbb{I}(s^k; s^k_{-\delta t})$ for key states exceeds that of normal states.

For clarity, consider the example of picking up a cube. In this scenario, a key state $s^k$ may denote the arm's end-effector firmly grasping the cube, while $s^k_{-\delta t}$ represents a state where the arm is nearing the cube, poised to initiate contact. Prior to the cube's secure grip, the robotic arm might approach it through diverse trajectories. However, states leading up to the cube being firmly grasped exhibit inherent similarities in the semantic latent space.

### A.3 TASK DETAILS

We selected four pre-trained VLMs, CLIP[1], BLIP[2], BLIP[3], and FLAVA[4], all of which can perform the image-text-matching task to generate compatible scores between instructions and visual states. The task instructions are generated by GPT4 based on our carefully crafted prompts, and visual states are rendered by the simulation environment. We obtained checkpoints of VLMs from their official repositories.

Our four tasks and pre-collected demonstrations are sourced from Maniskill2[5]. All tasks operate within an action space featuring 8 Degrees of Freedom (DoF) for position-based control. Notably, the observation space, which also serves as the state space, exhibits variations contingent upon the specific task. For instance, in the StackCube task, 25 dimensions of the observation space represent the agent's proprioceptive signal, including robot joint positions, joint velocities, as well as the position and pose of the robot base. The additional 30 dimensions convey external environmental signals, detailing the position and pose of the Tool Center Point (TCP), cubeA, and cubeB, along with their relative positions to each other. For the PegInsertionSide task, while the agent's proprioceptive signals remain consistent, there's a distinct shift in the 25-dimensional environmental signal. Specifically, this signal encompasses the position and pose of the Tool Center Point, the peg's position and pose (which may closely coincide with the TCP pose upon being grasped by the robotic arm), the peg's size, the target hole's position, and its size or radius. In the TurnFaucet experiment, given the diverse geometric shapes of the faucets, we incorporate an extra three-dimensional informational input, enabling the policy network to differentiate among the faucets. For a comprehensive insight, we recommend readers refer to the official documentation.

- Pick and Place Cube: This task requires a robot to pick up a cube and accurately place it within 2.5 cm of a specified 3D goal position while remaining stationary.

- Stack Cube: This task requires a robot to pick up a red cube and stably place it on top of a green cube without holding onto it.

- Peg Insertion Side: This task requires a robot to insert a peg halfway into a horizontal hole in a box under varying initial conditions.

- Turn Faucet: This task involves a robot rotating a faucet handle to a specified angular distance, with variations in initial conditions and faucet models for training and testing.

---

[1] https://github.com/openai/CLIP
[2] https://github.com/salesforce/BLIP
[3] https://github.com/salesforce/LAVIS/tree/main/projects/blip2
[4] https://huggingface.co/facebook/flava-full
[5] https://github.com/haosulab/ManiSkill2

### A.4 NORMALIZATION STRATEGIES

To simplify notation and model output structure, we propose normalizing trajectories to a consistent length, followed by a proportional rescaling during control policy training. Three normalization approaches are suggested:

- Truncate all trajectories to the minimal length.

- Pad trajectories to the maximal length.

- Interpolate trajectories to the minimal length.

Empirical tests show that the first approach is viable when trajectory length disparity is minor. Still, it compromises information for trajectories with substantial length differences, disproportionately localizing key states in the early segments. For the second approach, Reflect Padding outperforms Zero Padding, because it avoids abrupt informational transitions in the trajectory. However, in practical applications, padding might cause key state localizations within the padded steps, thereby hindering effective planning during the manipulation policy training. For the third approach, interpolating to the maximal length is feasible but incurs additional computational costs. Hence, in our experiments, we adopted interpolation to the minimal length for normalization.

### A.5 KEY STATE LOCALIZATION NETWORK TRAINING

In training the Key State Localization Network, the objective is to extract key states with crucial step information. Given this unique goal, the task neither requires a test set nor raises concerns regarding overfitting. Thus, all 1,000 trajectories from the dataset were utilized for training the Key State Localization Network. We applied normalization preprocessing to the input trajectory length of the network, scaling it to match the minimal trajectory length within the dataset. The batch size is set at 20. Analogous to policy training, the Adam optimizer is employed with an initial learning rate of $5 \times 10^{-4}$. The learning rate undergoes cosine annealing as the iteration count rises. We train the network 200 epochs. To prevent the network from converging to local optima, we introduce a weight decay of $1 \times 10^{-3}$ and a dropout rate of 0.3. Experimentally, we observed potential conflicts when concurrently optimizing entropy and chronological loss. To address this, we introduce an alternating optimization strategy, wherein during the early training phase, the entropy term and the chronological loss are not concurrently included in the overall objective function. This approach mitigates potential conflicts between the two losses, preventing convergence to local optima.

### A.6 CONTROL POLICY TRAINING

**Overview:** A sequential decision-making problem can be defined as a discrete-time finite Markov decision process by a 7-tuple $M = (\mathcal{S}, \mathcal{A}, \mathcal{P}, r, \rho_0, \gamma, H)$, where $\mathcal{S}$ is a state set, $\mathcal{A}$ is an action set, $\mathcal{P} : \mathcal{S} \times \mathcal{A} \times \mathcal{S} \to \mathbb{R}_+$ is a transition probability distribution, $\gamma \in [0, 1]$ is a discount factor, and $H$ is the horizon of the process. For imitation learning problems, a complete state-action trajectory can be defined as $\tau = (s_0, a_0, ..., s_t, a_t)$, where $s_0 \sim \rho_0(s_0)$, $a_t \sim \pi_\theta(\cdot|s_t)$, and $s_{t+1} \sim \mathcal{P}(\cdot|s_t, a_t)$. The training objective is to find an optimal policy $\pi_\theta$ that maximizes the expected discounted reward $\eta(\pi_\theta) = \mathbb{E}_\tau[\sum_{t=0}^{T} \gamma^t r(s_t, a_t, s_{t+1})]$ within the decision process. In our behavior cloning setting, the object manipulation tasks are characterized by their multi-stage complexity. Therefore, we relax the Markovian assumption to a non-Markovian setting, incorporating information beyond locality. Consequently, the policy can be expressed as $\pi_\theta(a_t|s_{t-T}, a_{t-T}, ..., s_t)$, leveraging information from states and actions with a fixed context length.

**Model architecture:** For a fair comparison, we adopt the similar transformer-based architecture as utilized in CoTPC (Jia et al., 2023) and the Decision Transformer (Chen et al., 2021). Notably, despite leveraging more key state information, our approach introduces no extra trainable parameters when compared to CoTPC. Initially, we employ two separate 2-layer MLPs to embed the histories of states and actions. Subsequently, we feed the state and action embedding tokens, and auxiliary latent key state tokens, into a transformer network, which operates based on causal attention and is augmented with positional embeddings. The intermediate features extracted by the hidden layer are processed through a lightweight MLP to predict key states, while the output features from the transformer are processed through another MLP-based predictor for action prediction. Thus, this

network enables concurrent action prediction and dynamic key state prediction. The total parameter count for this network approximates 1 million.

**Training details:**  To ensure reproducibility and reliability, we use the same random seed for all tasks (defaulted to 0). During training, we randomly sample 500 trajectories for all tasks. For testing, we not only test on trajectories seen during training but also on unseen trajectories. During training, we maintain an input context length of 60 for the policy network, with a batch size of 256. We use the Adam optimizer with an initial learning rate of $5 \times 10^{-4}$, decaying it using cosine annealing as the number of iterations increases. The total number of iterations is 2M. Different tasks have varying optimization difficulties. For more complex manipulation tasks, such as PegInsertionSide, we use a weight decay of $1 \times 10^{-3}$ and a dropout of $1 \times 10^{-2}$ to enhance generalization capability. We will release our code and provide more technical details later.

### A.7 KEY STATE GROUNDING VISUALIZATION

Fig. 8 displays the time-averaged images for these four tasks. In this figure, the first row shows the initial state of a randomly sampled demonstration for each task, while the second row presents the time-averaged images of this randomly selected demonstration. The third row depicts the time-averaged images after image enhancement. We can observe that for these four tasks, the overlap image is sharper when the current state is proximate to the key state. For the 'Turn Faucet' task, this observation might not be as pronounced as in the other three tasks. This can be attributed to the longer horizon of the demonstration for the 'Turn Faucet', meaning there are more images within a single demonstration, thus diluting the impact of each key state on the time-averaged images.

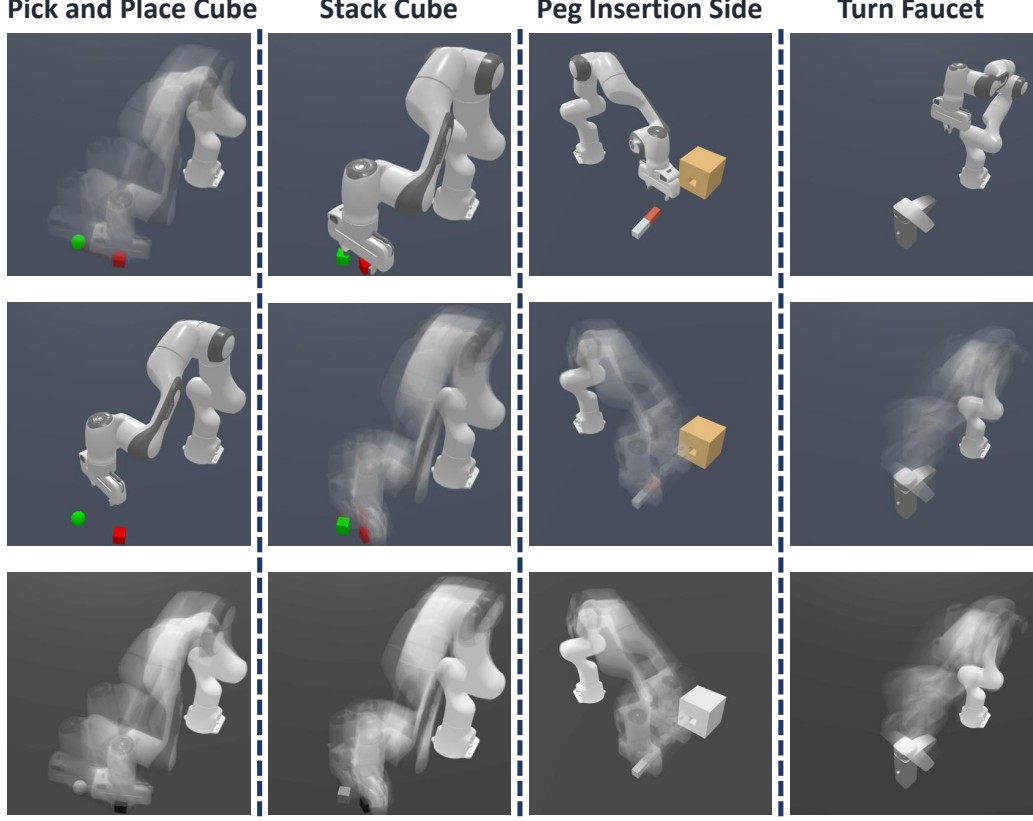

Figure 8: For all four tasks, we visualize the initial state of a randomly sampled demonstration, the time-averaged images, and the enhanced time-averaged images.

We infer key state localization network to obtain grounded key states, as shown in Fig. 9. It can be found that grounded key states not only satisfy the chronological order, that is, they are closer to the task completion step by step, but also exhibit some necessary states for specific task completion, such as the robot arm grasps the peg and the hole at the level of peg alignment in Peg Insertion Side task.

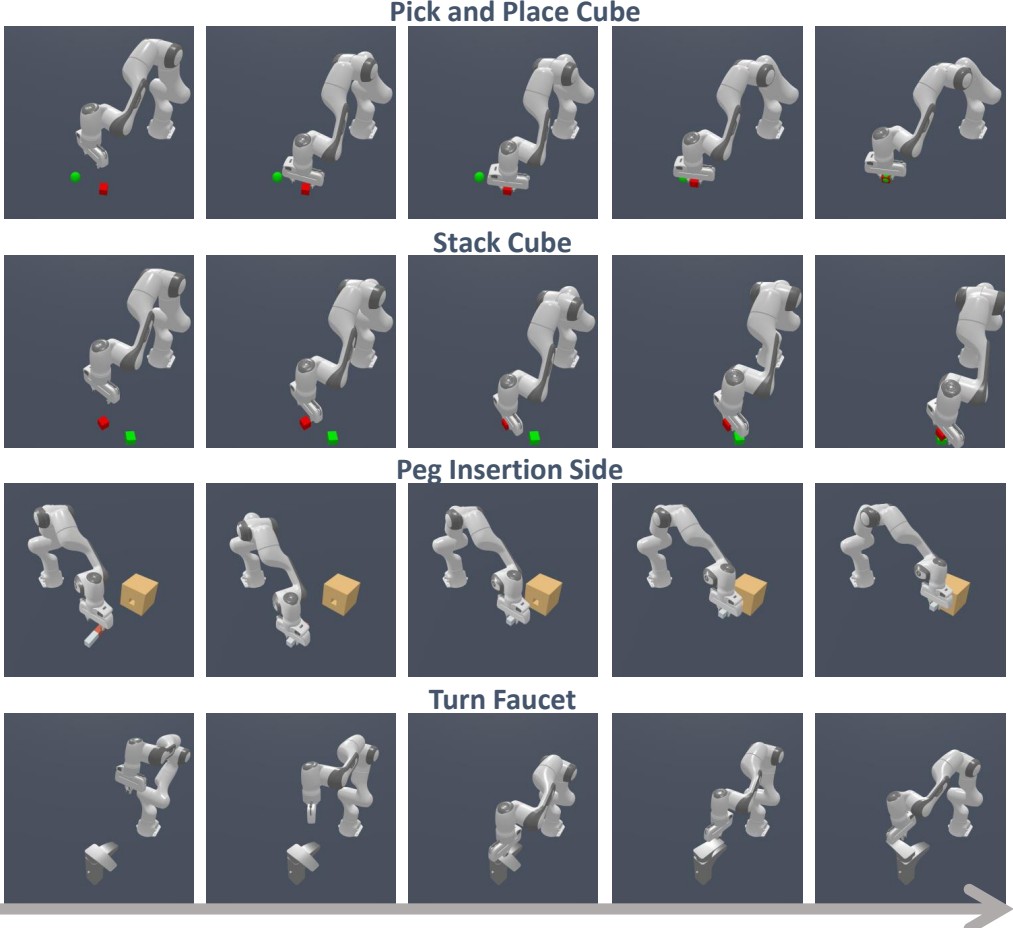

Figure 9: For four tasks from Maniskill2, we visualized the key states obtained when the number of key states grounded is set to five.

## A.8  LIMITATIONS AND FUTURE WORK

Although we show that our method can ground effective key states to guide the agent to complete the task, it still has some limitations. First of all, the now powerful LLMs are pure text-generation models, so they cannot clearly perceive the state space of the task to be completed, so the generated instructions with physical semantics may have a slight deviation from the actual task. We have also tested some multimodal models, but due to the scale of the parameters, they do not perform well in task planning. Second, we now design templates for different tasks that require a little bit of manual work. We hope that in the future there will be ways to automatically design templates to prompt LLM and automatically determine the optimal number of key states based on the task. Third, the ability of the foundation model will have some impact on our performance. For example, fine-tuning VLM can bring better grounding results, but annotated image-text pairs are also needed. How to bootstrap foundation models and avoid catastrophic forgetting are also promising future research directions.

A.9 PROMPTING EXAMPLES

First, we obtain descriptions of the key steps for each task by prompting GPT-4.

---

**Prompt:** Suppose you are a task planner. The task objective is that pick up a red cube and place it onto a green one. The success metric is that The red cube is placed on top of the green one stably and it is not grasped. Please give me 5 intermediate steps that are required to achieve this task.

Remember:

1. The intermediate steps should be critical and necessary.
2. The intermediate steps should be static physical states, not dynamic ones.
3. The intermediate step is visually recognizable, not just an abstract concept.
4. The description should be general not too detailed.
5. The length of each subgoal is less than 20 words.
6. Put the output in the following JSON structure {1: [text], 2: [text] ... }. [text] are the intermediate steps to be filled.

Exemplar:

For a task that picks up a cube and moves it to a goal position, the output can be as follows:

```
{    'Step 1': 'Robotic arm positioned above the cube.',
     'Step 2': 'Robotic arm's gripper aligned with the cube.',
     'Step 3': 'Robotic arm has securely grasped the cube.',
     'Step 4': 'Robotic arm lifted, holding the cube.',
     'Step 5': 'Robotic arm positioned above the goal position.'
}
```

---

**Response:** Certainly! Here are the 5 intermediate steps for the task objective of picking up a red cube and placing it onto a green one:

```
{
     'Step 1': 'Robotic arm positioned above the cube.',
     'Step 2': 'Robotic arm's gripper aligned with the cube.',
     'Step 3': 'Robotic arm has securely grasped the cube.',
     'Step 4': 'Robotic arm positioned above the green cube.',
     'Step 5': 'Red cube is lowered onto the green cube but not yet
         released.'
}
```

These steps fulfill the criteria for being critical and necessary, visually recognizable, and are static physical states. They are also general and each less than 20 words long.

---

**Prompt:** Suppose you are a task planner. The task objective is that the robotic arm inserts a peg into the horizontal hole in a box. The success metric is that half of the peg is inserted into the hole. Please give me 5 intermediate steps that are required to achieve this task.

Remember:

1. The intermediate steps should be critical and necessary.
2. The intermediate steps should be static physical states, not dynamic ones.
3. The intermediate step is visually recognizable, not just an abstract concept.
4. The description should be general not too detailed.
5. The length of each subgoal is less than 20 words.
6. Put the output in the following JSON structure {1: [text], 2: [text] ... }. [text] are the intermediate steps to be filled.

Exemplar:

For a task that picks up a cube and moves it to a goal position, the output can be as follows:

```
{    'Step 1': 'Robotic arm positioned above the cube.',
     'Step 2': 'Robotic arm's gripper aligned with the cube.',
     'Step 3': 'Robotic arm has securely grasped the cube.',
     'Step 4': 'Robotic arm lifted, holding the cube.',
     'Step 5': 'Robotic arm positioned above the goal position.'
}
```

**Response:**
```
{
     "Step 1": "Robotic arm positioned above the peg.",
     "Step 2": "Robotic arm's gripper aligned with the peg.",
     "Step 3": "Robotic arm has securely grasped the peg.",
     "Step 4": "Peg tip aligned with the hole's opening.",
     "Step 5": "Peg is partially inserted into the hole."
}
```

Next, in order to reduce the noise in the key state descriptions, we use GPT-4 to rephrase each text description.

**Prompt:** Given a set of intermediate steps as follows, regenerate 10 different descriptions for each intermediate step.

```
{    'Step 1': 'Robotic arm positioned above the cube.',
     'Step 2': 'Robotic arm's gripper aligned with the cube.',
     'Step 3': 'Robotic arm has securely grasped the cube.',
     'Step 4': 'Robotic arm lifted, holding the cube.',
     'Step 5': 'Robotic arm positioned above the goal position.'
}
```

**Response:** Certainly! Here are 10 different descriptions for each of your provided intermediate steps:
```
{
  "Step 1": [
    "Arm aligned over the red cube.",
    "Arm directly above target cube.",
    "Robotic arm hovers over red cube.",
    "End effector above the red cube.",
    "Arm set over the designated red cube.",
    "Arm poised above red cube.",
    "Arm stationed over red cube.",
    "Robot arm aimed above red cube.",
    "Vertical alignment with red cube achieved.",
    "Robotic arm's tip above red cube."
  ],
  "Step 2": [
    "Gripper oriented to match cube's position.",
    "Claw alignment synced with cube.",
    "Gripper set in line with red cube.",
    "Gripper ready for cube grasp.",
    "Claw faces cube directly.",
    "Gripper targeted toward the cube.",
    "Claw and cube in alignment.",
    "Gripper square with the red cube.",
    "Gripper aiming at the cube.",
    "End effector lined up with cube."
  ],
  "Step 3": [
    "Cube firmly in the gripper's hold.",
```

```
      "Gripper clenched, red cube secured.",
      "Gripper tightly encloses red cube.",
      "Claw locked, holding red cube.",
      "Solid grasp on red cube confirmed.",
      "Cube captured in gripper's jaws.",
      "Red cube tightly gripped.",
      "Gripper engaged, cube in possession.",
      "Red cube securely held by claw.",
      "Firm grip on red cube established."
  ],
  "Step 4": [
      "Arm raised, cube securely in tow.",
      "Cube elevated by lifted arm.",
      "Arm ascended, holding red cube.",
      "Cube hoisted by elevated arm.",
      "Red cube lifted off the ground.",
      "Arm's height adjusted, cube in hand.",
      "Arm and cube raised together.",
      "Elevated arm holds red cube.",
      "Cube in the air, arm lifted.",
      "Arm elevated, cube securely grasped."
  ],
  "Step 5": [
      "Arm centered over green target.",
      "Arm hovers above green dot.",
      "End effector above green goal.",
      "Arm set above green destination.",
      "Arm aimed at the green goal area.",
      "Robot arm stationed above green mark.",
      "Arm directly above the green objective.",
      "Poised arm over green target spot.",
      "Arm's tip over the green goal.",
      "Aligned arm above green position."
  ]
}
```

Each set contains 10 alternate descriptions for the individual steps that are static, visually recognizable, and meet your word-length requirement.

---

**Prompt:** Given a set of intermediate steps as follows, regenerate 10 different descriptions for each intermediate step.

```
{
    "Step 1": "Robotic arm positioned near the faucet.",
    "Step 2": "Robotic arm's gripper aligned with faucet handle.",
    "Step 3": "Robotic arm has securely grasped the handle.",
    "Step 4": "Robotic arm is in position to rotate the handle.",
    "Step 5": "Handle turned to target angular distance."
}
```

---

**Response:**

```
{
    "Step 1": [
        "The robotic arm is located above the peg.",
        "Above the peg, the robotic arm is positioned.",
        "The peg is beneath the robotic arm.",
        "The robotic arm is in place over the peg.",
        "Position of the robotic arm: above the peg.",
        "The peg lies below the positioned robotic arm.",
```

```
        "Robotic arm is situated over the peg.",
        "Above the peg, the robotic arm takes its position.",
        "The robotic arm hovers above the peg.",
        "Robotic arm is set above the peg."
    ],
    "Step 2": [
        "The gripper on the robotic arm is aligned with the peg.",
        "Alignment achieved between gripper and peg.",
        "Robotic arm's gripper lines up with the peg.",
        "The peg and the robotic arm's gripper are in alignment.",
        "Robotic arm's gripper is set to align with the peg.",
        "The peg and the gripper of the robotic arm are aligned.",
        "Robotic arm's gripper is in line with the peg.",
        "Gripper and peg are in a straight line.",
        "Alignment between the gripper and peg is complete.",
        "The robotic arm's gripper and the peg line up."
    ],
    "Step 3": [
        "The peg is securely held by the robotic arm.",
        "Robotic arm has a firm grasp on the peg.",
        "Securely, the robotic arm grasps the peg.",
        "The peg is in the robotic arm's secure grip.",
        "Robotic arm maintains a secure hold on the peg.",
        "The peg has been securely gripped by the robotic arm.",
        "The robotic arm's grip on the peg is secure.",
        "The peg is firmly in the grasp of the robotic arm.",
        "Robotic arm has the peg in a secure grasp.",
        "A secure hold on the peg is achieved by the robotic arm."
    ],
    "Step 4": [
        "Peg tip is in alignment with the hole opening.",
        "Tip of peg lines up with the hole.",
        "The peg's tip and the hole's opening are aligned.",
        "The peg tip is situated to align with the hole.",
        "Peg tip and hole opening are in a straight line.",
        "Alignment between peg tip and hole is achieved.",
        "Tip of the peg is lined up with the hole.",
        "Peg tip is set to align with hole's opening.",
        "Peg and hole are aligned at the tip.",
        "The peg tip is in line with the hole."
    ],
    "Step 5": [
        "Partial insertion of the peg into the hole is achieved.",
        "Peg is half-way into the hole.",
        "Peg has been partially put into the hole.",
        "Half of the peg is in the hole.",
        "Peg is semi-inserted into the hole.",
        "Part of the peg is inside the hole.",
        "The peg is not fully, but partially inserted.",
        "Peg is in the hole up to its midpoint.",
        "A portion of the peg is inserted into the hole.",
        "Some of the peg is within the hole."
    ]
}
```

**Prompt:** Given a set of intermediate steps as follows, regenerate 10 different descriptions for each intermediate step.

```
{
```

```
    "Step 1": "Robotic arm positioned above the peg.",
    "Step 2": "Robotic arm's gripper aligned with the peg.",
    "Step 3": "Robotic arm has securely grasped the peg.",
    "Step 4": "Peg tip aligned with the hole's opening.",
    "Step 5": "Peg is partially inserted into the hole."
}
```

**Response:**

```
{
    "Step 1": [
        "The robotic arm is situated near the faucet.",
        "Close to the faucet, the robotic arm is positioned.",
        "Robotic arm is placed close to the faucet.",
        "The faucet is adjacent to the positioned robotic arm.",
        "Robotic arm is located in proximity to the faucet.",
        "Robotic arm is nearby the faucet.",
        "Position of the robotic arm: near the faucet.",
        "Robotic arm hovers near the faucet.",
        "The robotic arm takes its position close to the faucet.",
        "The robotic arm is set near the faucet."
    ],
    "Step 2": [
        "The gripper on the robotic arm aligns with the faucet
            handle.",
        "Robotic arm's gripper lines up with the handle.",
        "Alignment between the robotic arm's gripper and faucet
            handle is achieved.",
        "Robotic arm's gripper is in line with the faucet handle
            .",
        "Gripper and faucet handle are in a straight line.",
        "Robotic arm's gripper and the handle are aligned.",
        "The faucet handle and the robotic arm's gripper are in
            alignment.",
        "The gripper of the robotic arm is set to align with the
            handle.",
        "Robotic arm's gripper is oriented towards the handle.",
        "The robotic arm's gripper and the faucet handle line up."
    ],
    "Step 3": [
        "The faucet handle is securely held by the robotic arm.",
        "Robotic arm has a firm grip on the faucet handle.",
        "The faucet handle is in the robotic arm's secure grip.",
        "Robotic arm maintains a secure hold on the faucet handle
            .",
        "The faucet handle is firmly in the grasp of the robotic
            arm.",
        "A secure hold on the faucet handle is achieved by the
            robotic arm.",
        "Robotic arm's grip on the faucet handle is secure.",
        "Securely, the robotic arm grasps the faucet handle.",
        "Robotic arm has the faucet handle in a secure grasp.",
        "The handle is securely gripped by the robotic arm."
    ],
    "Step 4": [
        "Robotic arm is ready for handle rotation.",
        "Robotic arm is set to rotate the faucet handle.",
        "The robotic arm is in the handle-rotating position.",
        "Robotic arm is in handle-rotating stance.",
        "Robotic arm prepared for handle rotation.",
```

```
        "Position for rotating the handle is established.",
        "Robotic arm aligns for handle rotation.",
        "Robotic arm in a position to turn the handle.",
        "The robotic arm is poised to rotate the handle.",
        "Robotic arm is lined up for handle rotation."
    ],
    "Step 5": [
        "Target angular distance of the handle is reached.",
        "Handle has been rotated to the specified angle.",
        "Faucet handle turned to the goal angle.",
        "Handle rotation reaches target angular distance.",
        "Handle has achieved the necessary angular rotation.",
        "Handle is at the target rotation angle.",
        "Faucet handle is turned past the designated angle.",
        "Handle is rotated to the required angle.",
        "Faucet handle meets target angular criteria.",
        "Handle successfully turned to desired angle."
    ]
}
```

