# OpenReview forum: "InfoGround: Ground Manipulation Concepts with Maximal Information Boost"
_ICLR.cc/2024/Conference — Submitted to ICLR 2024_

### Official Review · Reviewer_dSMV · 2023-10-31

**Soundness:** 3 good
**Presentation:** 4 excellent
**Contribution:** 3 good
**Rating:** 6
**Confidence:** 5

**Summary:**

This paper proposes Mutual Information Boost (MIB) to generate key states for a task specified by natural language. Based on the observation that the uncertainty of a state reduces rapidly when approaching a key state, the proposed method identifies key states by maximizing the rate of mutual information changes between the key state and its preceding states. The paper showed that the selected subgoals can guide the policy to reach a higher success rate in manipulation tasks.

**Strengths:**

- The paper identifies the ambiguity issue of using VLMs to ground key states and proposes a method that uses changes in mutual information to improve it.
- The information boost can apply to any key states that are defined by physical constraints and is a more principled way to identify key states.
- The paper presents the method and idea clearly, and it is easy to follow.

**Weaknesses:**

- The key states for some tasks are hard to describe using sentences, for example, in the task “hold the tube and align it with the table”, it is hard to describe the key state for “align” if not know the dimension, size, or relative orientation/positions for the tube and table. The proposed method would be more practical if it could identify key states without the state-by-state instruction generated by LLMs.
- Most tasks in the experiment do not really need to understand key states. For example, tasks such as pick and place cubes can be learned with pure behavior cloning (it is a bit surprising the performance reported in this paper was not that good), but it is harder to see the effect of key states grounding with short-horizon pick-and-place experiments.
- There is no qualitative example of the identified key states. It is hard to understand if the selected key states are good and why predefined rules are worse than MIB.

**Questions:**

- If the key states have the maximum mutual information boost, why there is a need to set “n” (the margin) to ensure any two key states are not too close to each other? It is possible that the two key states are quite close, for example, for the task “inserting the key to the keyhole, aligning the key with the keyhole, and moving the key into the keyhole are both key states and can be quite close to each other temporally.
- How much does paraphrase instruction help? If the embedding doesn’t have knowledge about the physically relevant description, changing the sentence wouldn’t have much help.

---

> ### Author Response · Authors · 2023-11-20
>
> Dear Reviewer dSMV,
>
> Thanks for the constructive comments and suggestions. We also appreciate your acknowledgment of the principledness of our proposed metric, as well as the quality of the presentation.
>
> We also thank you for many insightful questions that promote discussions to further reveal the scientific merits of the proposed metric. Please find our answers and clarifications below in detail.
>
> We hope the answers can help finalize your assessment and the rating of our paper. Please also let us know if you have any further questions that we need to provide additional clarifications.
> ***
> **W1: The key states for some tasks are hard to describe using sentences, for example, in the task “hold the tube and align it with the table”, it is hard to describe the key state for “align” if not know the dimension, size, or relative orientation/positions for the tube and table. The proposed method would be more practical if it could identify key states without the state-by-state instruction generated by LLMs.**
>
> **A**: Thanks for the insightful comment. We agree that it would be nice if the agent could automatically discover key states for policy learning. And this is actually the direction we are heading to. In the sense that, manually defined key states (e.g., the one obtained by predefined rules in CoTPC) are too limited in terms of diversity.
>
> For example, some key states are semantically meaningful but hard to be described by a set of predefined rules. We alleviate this constraint by employing LLMs to propose the key states (instead of constructing more rules). We admit that LLMs have their own limitations, which is out of the scope of our current paper. We focus on the grounding in this work but will study the limitation of LLMs in proposing key states in future research.
>
> However, we would also like to point out that there is a benefit of leveraging LLMs for proposing the states, as now the meaning of the key states is aligned with human understanding, which can help promote transparency of the grounded key states and the explainability of the trained policies.
>
>  ***
> **W2: Most tasks in the experiment do not really need to understand key states. For example, tasks such as pick and place cubes can be learned with pure behavior cloning (it is a bit surprising the performance reported in this paper was not that good), but it is harder to see the effect of key states grounding with short-horizon pick-and-place experiments.**
>
> **A**: Thanks for the question. Even though Pick & Place Cube seems to be a short-horizon task, our experiments still show that key states help policy learning.
>
> The comparison (on Pick & Place Cube) between the baselines not using key states and CoTPC already demonstrates the effectiveness of the key states. Also, the comparison between CoTPC and our results shows that there are even more key states that are useful even within such a short-horizon P&P Cube task.
>
> For example, in CoPTC, only one key state is defined, which is “grasp.” However, our method employs an LLM to propose many more (semantically meaningful) key states and ground them with the proposed information-theoretic criterion. For example, the key states proposed for Pick & Place Cube in our method are:
>
> {
> - `KS 1`: Robotic arm positioned above the cube.
> - `KS 2`: Robotic arm's gripper aligned with the cube.
> - `KS 3`: Robotic arm has securely grasped the cube.
> - `KS 4`: Robotic arm lifted, holding the cube.
> - `KS 5`: Robotic arm positioned above the goal position.
>
> }
>
> The flexibility of employing LLMs to propose more semantically meaningful key states enables the better performance of our trained policies. It is also evidence that short-horizon tasks can also benefit from fine-grained (but semantically meaningful) key states.
>
> Additionally, we also conducted experiments on the more complex long-horizon task, Franka Kitchen, to verify the effectiveness of key state grounding. The basic neural network architecture is Decision Transformer (DT). The results are shown below. We can find that in this task, DT with key states can achieve superior performance over the one without key states. In the unseen environment, the relative improvement was as high as 73.7%.
> | Franka Kitchen (Success rate) | Seen environment | Unseen environment |
> |-------------------------------|------------------|--------------------|
> | Without key states            | 78.4             | 13.7               |
> | With key states               | 81.8             | 23.8               |
>
>
> ***

---

> > ### Author Response · Authors · 2023-11-20
> >
> > **W3: There is no qualitative example of the identified key states. It is hard to understand if the selected key states are good and why predefined rules are worse than MIB.**
> >
> > **A**: In Appendix A.7, we have presented some qualitative examples of identified key states.
> >
> > The reason for the selected key states to be better than the predefined rules mainly lies in the fact that the predefined rules can only give a very small number of key states for training the policies. For example, for the simple Pick & Place Cube task, the predefined rule only gives one key state "grasp" due to the difficulty of manually specifying the rules.
> >
> > However, leveraging LLMs, our method can propose diverse (semantically meaningful) key states, which are then grounded for learning policies. For the P&P Cube task, there are, in contrast, five key states proposed, which are described in W2. The selected key states result in better policies due to their diversity and semantical meaningfulness.
> >
> >
> > In summary, our method's advantage is its flexibility and autonomy in grounding more effective key states instead of relying on predefined heuristic rules. Human predefined labels are fixed and require extensive human labor to annotate key states, leading to a limited number of annotated key states as well as inferior policy performance.
> >
> > ***
> > **Q1: If the key states have the maximum mutual information boost, why there is a need to set “n” (the margin) to ensure any two key states are not too close to each other? It is possible that the two key states are quite close, for example, for the task “inserting the key to the keyhole, aligning the key with the keyhole, and moving the key into the keyhole are both key states and can be quite close to each other temporally.**
> >
> > **A**: Thank you for your insightful observation. In fact, in our experiments, we typically set "n" to 1 or 2, which are relatively small values. The primary significance of introducing "n" is to ensure the chronological correctness of key state grounding, rather than to see it as a relatively large margin to prevent key states from being too close to each other.
> >
> > Our experiments have shown that setting "n" to 1 or 2 has minimal variation in the obtained key states and that using these values can stabilize the training process, resulting in faster convergence. In our experience, setting n to 1 or 2 speeds up convergence by about 30%.
> > ***
> > **Q2: How much does paraphrase instruction help? If the embedding doesn’t have knowledge about the physically relevant description, changing the sentence wouldn’t have much help.**
> >
> > **A**: This operation is observed to stabilize the training (optimization) process. In our experience, the incorporation of paraphrasing has been observed to speed up the training convergence by 20%.
> >
> > The necessity of paraphrases lies in the fact that pretrained visual-language encoders may be subject to domain gaps and insufficient granularity in their training data and thus introduce noise in the compatibility function (between the image and the key state description), which further induces learning instability. Therefore, the primary purpose of paraphrasing the instruction is to create semantically equivalent versions of the key state description and use them to form an averaged compatibility function, so that the noise in the encoders can be canceled out.

---

> ### Author Response · Authors · 2023-11-22
>
> Dear Reviewer dSMV,
>
> As time passes by, could you take a look at our rebuttal and let us know whether it resolves your concerns? We are also here to answer any further questions you may have.
>
> Best,
>
> The Authors

---

> > ### Comment · Reviewer_dSMV · 2023-11-23
> > **Thanks for the response!**
> >
> > I would like to thank the authors for additional discussions and results.
> >
> > I agree that the LLM proposed key states can provide the explanation and ground the key states. However, as some key states cannot be described using texts, it is not realistic to mainly rely on descriptions to identify key states. The experiments can use the proposed key state descriptions from LLMs because the tasks are relatively easy. If we change the peg to be longer or a different geometry, the same LLM proposed key state "positioned above the peg" no longer provides useful information as the robot needs to find a specific grasping pose to keep balance. In this case, the infoboost loss is more important than the compatibility loss. It will be helpful to show how infoboost loss contributes to the performance so we can understand how we can ground the key states even when the sentence description is not that informative.

---

> > > ### Author Response · Authors · 2023-11-23
> > >
> > > Dear Reviewer dSMV,
> > >
> > > Thanks for your insightful comment!
> > >
> > >
> > > We agree that leveraging LLM to propose manipulation concepts would be noisy since using language to describe physical states is limited. And our goal here is to alleviate the noise and provide better grounding.
> > >
> > >
> > > However, we would like to clarify that we are not relying on LLMs to perform the actual manipulation (e.g., predicting the actions), which requires an accurate perception of the environment, which current LLMs lack. In contrast, the proposed concepts, even though very coarse (like “positioned above the peg”), can be grounded via the proposed method to specific and accurate physical states from the demonstrations. With this, we like to say that our method comes with a certain level of robustness to the capability of LLMs (in proposing more detailed instructions), in the sense that our method is orthogonal to directly applying LLM for manipulation.
> > >
> > >
> > > To verify the robustness of our method, we have performed an ablation study with respect to different LLM- or VLM-based compatibility functions (e.g., different descriptions and encoding of the descriptions that simulate a large variation). The results are in Table 4 of the paper, and are also displayed below:
> > >
> > > | METHODS    | P&P CUBE SEEN | P&P CUBE UNSEEN | STACK CUBE SEEN | STACK CUBE UNSEEN | TURN FAUCET SEEN | TURN FAUCET UNSEEN | PEG INSERTION SEEN | PEG INSERTION UNSEEN |
> > > |------------|---------------|-----------------|-----------------|-------------------|------------------|--------------------|--------------------|----------------------|
> > > | CLIP   | 73.2   | 65.0	| 66.0   | 51.0	| 56.0   | 40.0    | 57.0   | 7.75	|
> > > | CLIP+MIB   | **78.8** | **71.0**  | **76.4** | **56.0**  |**61.0** | **60.0**  | **75.0** | **21.5**   |
> > > | BLIP   | 69.0   | 64.0	| 52.8   | 26.0	| 53.8   | 44.0    | 52.6   | 11.3	|
> > > | BLIP+MIB   | **85.0** | **73.0**  | **68.8**   | **50.0**    | **61.6**   |**59.0**	| **71.6** | **20.0**	|
> > > | BLIP2  | 73.0   | 59.0	| 56.6   | 43.0	| 54.6   | 51.0    | 59.4   | 11.3	|
> > > | BLIP2+MIB  | **76.8**   |**68.0**| **70.0** | **52.0**	| **56.4**   | **57.0**	| **70.8**   | **20.8**    |
> > > | FLAVA  | 74.0   | 59.0	| 40.2   | 28.0	| 55.0   | 56.0    | 63.6   | 13.0	|
> > > | FLAVA+MIB  | **83.0** | **73.0**  | **72.0** | **54.0**  | **65.0** | **60.0**  | **69.0** | **18.3**|
> > >
> > >
> > >
> > >
> > > The results demonstrate that the proposed MaxInfoBoost (MIB) criterion can improve over different compatibilities consistently, showing the effectiveness of the proposed, and evidencing that LLM is not the key to the success of the proposed criterion.
> > >
> > >
> > > Specifically, since both the Turn Faucet and Peg Insertion tasks feature variable object geometries, we can see that for these two tasks, especially for the Peg Insertion task, the introduction of MIB loss can improve the performance on both seen and unseen environments. Especially,  in the unseen environment, the relative performance improvement is substantial.
> > >
> > >
> > > Since manipulation concepts are important for efficient learning and generalization of interaction policies, the contribution of our work mainly lies in using LLMs to propose more fine-grained ones (compared to human-labeled ones) that can help with policy learning, while minimizing the annotation effort (of grounded states). Our experiments verify that the proposed MIB criterion is effective on various tasks from ManiSkill2. But we agree that fully relying on LLMs is not healthy, which is something we are not proposing here. Again, we just want to show that there is an alternative to human labeling of those grounded states, and we have verified its effectiveness.
> > >
> > >
> > > Thanks again for your valuable feedback! Please let us know any comments that you see helpful for improving our manuscript.

---

### Official Review · Reviewer_4ns7 · 2023-10-31

**Soundness:** 3 good
**Presentation:** 2 fair
**Contribution:** 2 fair
**Rating:** 5
**Confidence:** 4

**Summary:**

The paper introduces a method to ground manipulation concepts generated by Large Language Models (LLMs) into physical states, facilitating the training of efficient manipulation policies. The authors observe that current methods, which rely on multimodal foundation models to localize key states, often lack accuracy and semantic consistency. To address this, they propose an information-theoretic criterion called Maximal Information Boost (MIB) to enhance grounding efficiency without requiring extensive fine-tuning. The approach is based on the observation that the uncertainty of a state diminishes rapidly as it approaches a key state.

**Strengths:**

This paper presents an interesting idea of using the maximal information boost (MIB) principle to align key state descriptions in language with image states in demonstration trajectories. The final goal of this approach is to enable better imitation learning from demonstration trajectories. The proposed approach is interesting in the sense that it introduces a novel metric for enforcing grounding.

**Weaknesses:**

There are three major weaknesses of the paper.

First, the writing of the paper is not very clear. For example, the majority of the paper describes how we can use MIB to segment trajectories, but there is no description of how this MIB-derived segmentation is actually used in training the policy network. I believe this part should be at least briefly described in the paper. (I found it in the supplementary material, but I would like to mention that reviewers are not subject to reading the full supplementary material).

Second, while I found the introduced metric novel and interesting, I did not fully get its intuition. In particular, is it just the case that the speed of the robot arm will be very low when it's close to a "key state?" Because the uncertainty is essentially defined as s_t | s_{t-dt}.

Third, since the contribution of the paper lies in segmenting trajectories (the policy part is directly following CoTPC, based on the supplementary description), there is little comparison between the segmentation method and other related approaches, such as

- Skill Induction and Planning with Latent Language https://arxiv.org/abs/2110.01517
- Learning from Trajectories via Subgoal Discovery https://proceedings.neurips.cc/paper_files/paper/2019/file/6f518c31f6baa365f55c38d11cc349d1-Paper.pdf
- CompILE: Compositional Imitation Learning and Execution https://arxiv.org/abs/1812.01483
- Identifying useful subgoals in reinforcement learning by local graph partitioning https://dl.acm.org/doi/10.1145/1102351.1102454

among many, many others. The authors should discuss, and compare with these algorithms.

Also, it seems that the proposed method has the limitation that each key state can only be visited once in a demonstration trajectory (due to the Chronological assumption). Therefore, it can't model task descriptions that require loops (e.g., scoop all sugar, possibly multiple rounds, to transfer them from one bowl into another bowl).

**Questions:**

If the paper is based on CoTPC and CoTPC uses predefined key states, why would the performance of this paper be better than CoTPC? Is it because CoTPC is using a smaller number of key states? or due to noises in the labels?

Is the variance computed in the pixel space? (eq 7)

---

> ### Author Response · Authors · 2023-11-20
>
> Dear Reviewer 4ns7,
>
> Thanks for the constructive feedback and suggestions. We appreciate that you found our proposed metric novel and interesting.
>
> We understand that more explanations and comparisons would help your re-evaluation of our paper. Especially, on the intuition of the proposed metric, as well as the comparison with more baselines. Please find our answers and clarifications below in detail.
>
> We hope the answers can help finalize your assessment and the rating of our paper. Please also let us know if you have any further questions that we need to provide additional clarifications.
>
> ***
> **W1: The writing of the paper is not very clear. For example, the majority of the paper describes how we can use MIB to segment trajectories, but there is no description of how this MIB-derived segmentation is actually used in training the policy network. I believe this part should be at least briefly described in the paper. (I found it in the supplementary material, but I would like to mention that reviewers are not subject to reading the full supplementary material).**
>
> **A**: We appreciate your careful review and constructive feedback. We placed the policy network training details in the supplementary material due to page restrictions in the main manuscript. We thought the experimental setup could serve a similar purpose, but apparently, it causes a gap in the information flow. In the revised version, we have added a **new subsection 2.5** to elaborate on the training of the policy networks.
> ***
> **W2: While I found the introduced metric novel and interesting, I did not fully get its intuition. In particular, is it just the case that the speed of the robot arm will be very low when it's close to a "key state?" Because the uncertainty is essentially defined as s_t | s_{t-dt}**
>
> **A**: We appreciate your observations and would like to clarify that a key state does not necessarily correlate with a very low speed of the robot arm; it's more about the reduction of uncertainty in the latent state space.
>
> An intuitive example is outlined in Appendix A.2. To clarify and avoid any confusion regarding speed, consider another example: We are in the source room aiming to get to the target room, which is connected solely by an open door. Regardless of our path within the source room, accessing the target room necessitates passing through this door. This action of passing through the door symbolizes a key state, which is independent of the speed we traverse it at—we could do so either quickly or slowly. In the semantic space, however, the uncertainty s_t​∣s_{t−dt}​ is significantly reduced.
>
> Moreover, we would like to point out that since mutual information is invariant under scaling, the proposed metric is not affected by the speed, e.g., multiplying the state with a scalar can change the speed but not the mutual information. Thus, the proposed information-theoretic criterion is more than just determining key state by speed.
>
> ***

---

> > ### Author Response · Authors · 2023-11-20
> >
> > **W3: Since the contribution of the paper lies in segmenting trajectories (the policy part is directly following CoTPC, based on the supplementary description), there is little comparison between the segmentation method and other related approaches, such as**
> >
> > A. Skill Induction and Planning with Latent Language https://arxiv.org/abs/2110.01517
> >
> > B. Learning from Trajectories via Subgoal Discovery https://proceedings.neurips.cc/paper_files/paper/2019/file/6f518c31f6baa365f55c38d11cc349d1-Paper.pdf
> >
> > C. CompILE: Compositional Imitation Learning and Execution https://arxiv.org/abs/1812.01483
> >
> > D. Identifying useful subgoals in reinforcement learning by local graph partitioning https://dl.acm.org/doi/10.1145/1102351.1102454
> >
> > **A**: Thank you for the literature provided. We have carefully read and investigated all the papers you mentioned, which have been insightful for our work. We will incorporate them in our discussion and related work. However, we also found that not all of them are directly comparable, for example:
> >
> > A. Skill Induction and Planning with Latent Language (Pratyusha Sharma, Antonio Torralba, Jacob Andreas): the method proposed in this work takes in a demonstration and a goal description and outputs a set of sub-goals achieved during the demonstration. It does give information of key states involved in the task, but does not provide further groundings of the key states or sub-goals. Thus we could not get a quantitative comparison with it.
> >
> > D. Identifying useful subgoals in reinforcement learning by local graph partitioning(Özgür Şimşek, Alicia P. Wolfe, Andrew G. Barto): This work utilizes a graph of nodes in discrete space to discover subgoals in reinforcement learning. However, the problem we are facing involves states in high-dimensional continuous space. Directly applying their method to our problem would require a reasonable discretization of the high-dimensional state space beforehand for a fair comparison, which is challenging due to the high state complexity of our task.
> >
> > Although the aforementioned two papers are not directly comparable to our work, we did compare our approach with a recent behavior cloning-based key state extraction method, "Waypoint-Based Imitation Learning for Robotic Manipulation" [1]. This paper introduces an Automatic Waypoint Extraction (AWE) technique that decomposes demonstrations using dynamic programming to identify waypoints. We have also compared with the referred methods B and C. The results are listed below.
> >
> > | Method            | P&P Cube | Stack Cube | Turn Faucet | Peg Insertion | Mean  |
> > |-------------------|----------|------------|-------------|---------------|-------|
> > | B. Subgoal Discovery [2] | 79.3     | 44.1       | 51.7        | 48.9          | 56.0  |
> > | C. CompILE [3]       | 76.4     | 59.7       | 55.6        | 45.8          | 59.4  |
> > | AWE [1]           | 77.8     | 59.8       | 47.0        | 55.6          | 60.1  |
> > | CoTPC             | 75.2     | 58.8       | 56.4        | 52.8          | 60.8  |
> > | Ours              | 89.4     | 83.2       | 67.6        | 75.4          | 78.9  |
> >
> > We observed that the Subgoal Discovery, CompILE and AWE yield competitive results with CoTPC, yet our method still achieves superior performance in more complex tasks compared to other methods.
> >
> > [1] Shi, Lucy Xiaoyang, et al. "Waypoint-based imitation learning for robotic manipulation." arXiv preprint arXiv:2307.14326 (2023).
> >
> > [2] Paul, Sujoy, Jeroen Vanbaar, and Amit Roy-Chowdhury. "Learning from trajectories via subgoal discovery." Advances in Neural Information Processing Systems 32 (2019).
> >
> > [3] Kipf, Thomas, et al. "Compile: Compositional imitation learning and execution." International Conference on Machine Learning. PMLR, 2019.
> > ***

---

> > > ### Author Response · Authors · 2023-11-20
> > >
> > > **W4: It seems that the proposed method has the limitation that each key state can only be visited once in a demonstration trajectory (due to the Chronological assumption). Therefore, it can't model task descriptions that require loops (e.g., scoop all sugar, possibly multiple rounds, to transfer them from one bowl into another bowl).**
> > >
> > > **A**: Thanks for the insightful question. The proposed grounding method aims at connecting LLM-generated key state descriptions with the physical states, while the understanding of the (repetitive) structure of the task is mainly dependent on the capacity of the LLM itself, which is currently not our focus.
> > >
> > > Actually, our InfoGround method doesn't confine a key state to a single occurrence within a demonstration trajectory. Provided a key state recurs in the trajectory, our method is able to identify it as a key state each time, by leveraging the maximal information boost detailed in Section 2.3.
> > >
> > > However, the proposed task is an interesting scenario to examine the limitation of using LLMs to propose key states useful for achieving a physical manipulation task.
> > >
> > > To proceed with the discussion, we have prompted GPT4 with the following:
> > >
> > > Prompt: I am a robotic arm, please describe the step-by-step instructions for scooping all sugar.
> > >
> > > GPT:
> > > 1. Position: Move the robotic arm above the container holding the sugar.
> > > 2. Extend: Lower the arm until the scoop or shovel attachment is inside the sugar.
> > > 3. Angle: Tilt the scoop to a 45-degree angle to ensure maximum sugar collection.
> > > 4. Scoop: Close the scoop or shovel attachment to collect sugar, making sure it is full.
> > > 5. Lift: Raise the robotic arm, ensuring the scoop remains level to prevent sugar spillage.
> > > 6. Transport: Move the arm to the desired location for dispensing the sugar.
> > > 7. Release: Open the scoop or shovel attachment, allowing the sugar to pour out.
> > > 8. Repeat: Continue steps 1-7 until all sugar has been scooped and transferred.
> > >
> > > From the output, we can still see that LLM has a good understanding of the repetitive structure of the task (step 8), and can propose meaningful key state descriptions. This observation should be evidence that the proposed pipeline can still work for the proposed task of scooping all sugar. However, since there are no existing demonstrations for training the grounding, we would like to check this aspect in the future after collecting more data regarding tasks with repetitive patterns.
> > >
> > >
> > > ***
> > > **Q1: If the paper is based on CoTPC and CoTPC uses predefined key states, why would the performance of this paper be better than CoTPC? Is it because CoTPC is using a smaller number of key states? or due to noises in the labels?**
> > >
> > > **A:**: Thanks for the question. The main problem with CoTPC’s manually defined ground truth is that it only captures very few key states due to the limited annotation bandwidth. For example, in CoTPC, the predefined key state for Pick & Place Cube is only “grasp,” which is determined by checking if the gripper is in contact with the cube.
> > >
> > > However, our method employs an LLM to propose many more (semantically meaningful) key states and ground them with the proposed information-theoretic criterion. For example, the key states proposed for Pick & Place Cube in our method are:
> > >
> > > {
> > > - `KS 1`: Robotic arm positioned above the cube.
> > > - `KS 2`: Robotic arm's gripper aligned with the cube.
> > > - `KS 3`: Robotic arm has securely grasped the cube.
> > > - `KS 4`: Robotic arm lifted, holding the cube.
> > > - `KS 5`: Robotic arm positioned above the goal position.
> > >
> > > }
> > >
> > >
> > > From the above, we can see that KS1, 2, 4, and 5 are all meaningful states, but are not present in CoTPC's predefined key states, due to the limited annotation bandwidth. Our grounding can effectively ground them to the physical states, avoiding manual annotation or pre-defining rules. Thus, our method is more flexible and can leverage more meaningful key states to enhance the policy training, resulting in better performance. Also, please note that, this does not mean that the more the number of the key states, the better the policy performance. The key states still need to be semantically meaningful. We elaborate in the following table that by simply setting all the states to be key states can actually hurt the performance of the Peg Insertion task.
> > > | Number of key states | CoTPC(3) | Ours(5) | All-State(130) |
> > > |----------------------|----------|---------|----------------|
> > > | Success rate         | 52.8     | 75.4    | 18.0           |
> > >
> > >
> > >  ***
> > > **Q2: Is the variance computed in the pixel space? (eq 7)**
> > >
> > > **A**: No, the variance is computed in the latent space (e.g., the embedding of the image with CLIP). For simplicity, we use s_t to denote both the physical state and its latent embedding, which we clarify in the paper.

---

> ### Author Response · Authors · 2023-11-22
>
> Dear Reviewer 4ns7,
>
> As time passes by, could you take a look at our rebuttal and let us know whether it resolves your concerns? We are also here to answer any further questions you may have.
>
> Best,
>
> The Authors

---

### Official Review · Reviewer_KD76 · 2023-11-08

**Soundness:** 3 good
**Presentation:** 3 good
**Contribution:** 2 fair
**Rating:** 5
**Confidence:** 4

**Summary:**

The paper proposes a method called Maximal Information Boost (MIB) to efficiently connect language models with physical states in robotic manipulation tasks. It also introduces Key State Localization Network (KSL-Net) for matching physical states to their descriptions and predicting the likelihood of correctness. The method improves semantic alignment, enhances performance, and generalizes manipulation policies across various tasks.

**Strengths:**

- Introduce the Maximal Information Boost (MIB) criterion to enhance grounding efficiency without requiring fine-tuning.
- Develop the Key State Localization Network (KSL-Net) to determine the probability of each state being the grounded key state.
- Enhance semantic compatibility and improve generalization in robotic tasks.

**Weaknesses:**

- The LLM lacks visual information or constrains about the environment, which restricts its ability to make geometry-based decisions like collision avoidance.
- It is necessary to manually create various paraphrases that are semantically consistent depending on the tasks.
- The paper does not address how well the proposed approach can generalize to unseen tasks.

**Questions:**

- Do you need to train a separate model for each task? Given the simulated environments are relatively simple, how can you justify that the model is not overfitting the environment?
- The environments presented in the paper are clear scenarios, involving only a limited number of objects relevant to specific tasks. I'm curious about how the approach would handle complex scenes with multiple objects and some of them are unrelated to the task, as this might introduce ambiguity.
- Do you still need to do additional planning in your setup to complete the task with key frames?

---

> ### Author Response · Authors · 2023-11-20
>
> Dear Reviewer KD76,
>
> Thank you for the constructive comments and suggestions. We also appreciate your acknowledgment of the soundness, presentation, and effectiveness of our work.
>
> We understand that clarifications on the usage of LLMs in our pipeline, the role of paraphrasing with LLMs, and discussion on generalization are needed for your re-evaluation of the proposed key state grounding method. We also provide an experiment on more complex scenes to demonstrate the effectiveness of our algorithm when facing distractions or ambiguities. Please find details below.
>
> We hope the answers can help finalize your assessment and the rating of our paper. Please also let us know if you have any further questions that we need to provide additional clarifications.
>
> ***
> **W1: The LLM lacks visual information or constrains about the environment, which restricts its ability to make geometry-based decisions like collision avoidance.**
>
> **A**: We agree that the LLM lacks visual information to make geometry-based decisions like collision avoidance. However, this is not a critical issue for our method, since we are not relying on LLM to make low-level control decisions.
>
> Our method only leverages an LLM to provide potential key states involved in achieving a task, which does not need accurate geometric information. Our method then focuses on grounding these key states in the demonstrations, which then provides the grounded physical states for better policy learning and generalization. The experiments have shown that by grounding the LLM-generated key states, the trained policies can surpass human-provided key states, evidencing the effectiveness of the proposed grounding method.
>
> Again, thanks for the comment. In the future, we would like to see how the key states provided by an enhanced LLM (e.g., GPT-4V) can help with the learned policy.
>  ***
> **W2: It is necessary to manually create various paraphrases that are semantically consistent depending on the tasks.**
>
> **A**: It appears there may be a slight misunderstanding. In our experiments, various paraphrases are generated automatically by LLM, eliminating the need for manual modifications. The first stage of the process, instruction generation for getting key state descriptions, relies on minimal task priors such as specified task description. The second stage, paraphrase generation, is solely dependent on the instructions generated in the first stage and does not require any task priors.
>
> The necessity of paraphrases lies in the fact that pretrained visual-language encoders may be subject to domain gaps and insufficient granularity in their training data and thus introduce noise in the compatibility function (between the image and the key state description). Therefore, the primary purpose of paraphrasing the instruction is to create semantically equivalent versions of the key state description and use them to form an averaged compatibility function, so that the noise in the encoders can be canceled out.
>
> This operation is observed to stabilize the training (optimization) process. In our experience, the incorporation of paraphrasing has been observed to speed up the training convergence by 20%.
>
> ***
> **W3: The paper does not address how well the proposed approach can generalize to unseen tasks.**
>
> **A**: In terms of the generalization of the policy on unseen environments, Table 2 in our paper presents the required evaluation. To elaborate, Table 1 represents the performance of our proposed method on seen task configurations, while Table 2 demonstrates its performance on unseen task configurations, showcasing our strong generalization capabilities of the trained policies under key state guidance.
>
> In terms of the generalization of the grounding network, we would like to point out that it is generalizable by nature. The full training procedure of the grounding network requires no human annotation of the ground-truth groundings. In other words, the training objective for the grounding network is unsupervised, which contains only the compatibility function, and the mutual information boost (MIB) for enhancing the physical meaningfulness of the grounding. Thus the pipeline can be trained in any situation without concerning the test-time generalization.
>
> ***

---

> > ### Author Response · Authors · 2023-11-20
> >
> > **Q1: Do you need to train a separate model for each task? Given the simulated environments are relatively simple, how can you justify that the model is not overfitting the environment?**
> >
> > **A**: Thanks for the question. It actually reveals a nice point of our grounding framework. It is true that we train a separate grounding network (the KSL-Net) for each task to get the correct groundings, and then use the groundings to train a policy network to perform the task.
> >
> > With the discussion on the generalization of the policy networks and the KSL-Net in W3, we would like to point out that, instead of generalization, underfitting is actually a problem if we do not use a separate network for different tasks to derive the accurate grounding.
> >
> > First, our experiments show that the global minima of the loss function for a task gives us the best grounding result for that task. Thus it is critical to ensure that the grounding loss for each task is fully optimized. However, unifying the optimization of multiple tasks by training a single network can suffer from under-optimization, i.e., the groundings after training for a certain task may not be the global minima, compared to the one obtained by training a separate KSL-Net using the corresponding loss of this task. In a nutshell, we would like to emphasize that overfitting a separate KSL-Net to the grounding objective of a task can ensure global optimality of the grounding and is critical for the quality of the obtained grounding.
> >
> >
> > ***
> > **Q2: The environments presented in the paper are clear scenarios, involving only a limited number of objects relevant to specific tasks. I'm curious about how the approach would handle complex scenes with multiple objects and some of them are unrelated to the task, as this might introduce ambiguity.**
> >
> > **A**: Thanks for the suggestion. We have conducted experiments on the Franka Kitchen environment, which is a complex robotic task/environment, where a scene involves many other objects that may introduce ambiguity. With the same decision transformer architecture, our method (with key states) has improved from the baseline(without key states) for both seen and unseen environments. Especially in the unseen environment, the relative boost is 73.7%. The results are listed below.
> > | Franka Kitchen (Success rate) | Seen environment | Unseen environment |
> > |-------------------------------|------------------|--------------------|
> > | Without key states            | 78.4             | 13.7               |
> > | With key states               | 81.8             | 23.8               |
> >
> > The results demonstrate the importance of leveraging key states in improving the training efficiency and the generalization capability of the policy. Moreover, it shows that our grounding network can generalize to more complex scenarios.
> >
> > ***
> > **Q3: Do you still need to do additional planning in your setup to complete the task with key frames?**
> >
> > **A**: To ensure a fair comparison, no additional planning is employed. The framework is end-to-end, with the policy network concurrently undertaking action prediction and dynamic key state estimation during the training phase. This streamlined approach facilitates easy comprehension and seamless integration with other imitation learning pipelines.

---

> ### Author Response · Authors · 2023-11-22
>
> Dear Reviewer KD76,
>
> As time passes by, could you take a look at our rebuttal and let us know whether it resolves your concerns? We are also here to answer any further questions you may have.
>
> Best,
>
> The Authors

---

> > ### Comment · Reviewer_KD76 · 2023-11-23
> >
> > Thank you for your detailed response. While I appreciate and respect your points, I still consider the paper borderline, considering the experiments on relatively easy tasks. I will maintain my current ranking and I believe it's best to leave the final decision to the collective judgment of the other reviewers.

---

> > > ### Author Response · Authors · 2023-11-23
> > >
> > > Dear Reviewer KD76,
> > >
> > > Thank you for the time and effort you have dedicated to reviewing our manuscript and for your constructive comments.
> > >
> > > We would like to highlight that our chosen tasks for the experiments, though they may appear straightforward, are significantly challenging in multi-stage, long-horizon contexts. For example, in our selected Peg Insertion task, the mere 2.5mm clearance between the peg and the hole in the box presents a substantial challenge. Our experiments have shown that other methods perform not well in familiar (seen) environments and are mostly unsuccessful in unfamiliar (unseen) environments. In contrast, our approach not only performs well in seen environments but also shows a marked improvement in generalization ability in unseen environments compared to the baseline.
> > >
> > > Moreover, in addition to initial experiments, we also extended our experiments to more complex tasks, such as [Franka Kitchen](https://robotics.farama.org/envs/franka_kitchen/franka_kitchen/). This task presents a complex, multi-task environment in a kitchen setting with several common household items. As demonstrated in this challenging environment, our method, which incorporates key states, shows much improvement over the baseline, which lacks key states, in both seen and unseen environments, while employing the same decision transformer architecture.
> > >
> > > Since manipulation concepts are important for efficient learning and generalization of interaction policies, the contribution of our work mainly lies in using LLMs to automatically propose more fine-grained ones (compared to human-labeled ones) that can help with policy learning, while minimizing the annotation effort (of grounded states). Our experiments verify that the proposed MIB criterion is effective on various tasks from ManiSkill2. But we agree that fully relying on LLMs is not healthy, which is something we are not proposing here. Again, we just want to show that there is an alternative to human labeling of those grounded states, and we have verified its effectiveness.
> > >
> > > Thanks again for your valuable feedback! Please let us know if there is any information we can provide to help your final assessment of our paper.

---

### Official Review · Reviewer_S4jr · 2023-11-10

**Soundness:** 3 good
**Presentation:** 3 good
**Contribution:** 2 fair
**Rating:** 3
**Confidence:** 4

**Summary:**

The authors note that existing techniques using multimodal foundation models for key state identification are often imprecise and semantically inconsistent. To improve this, the authors introduce a concept named Maximal Information Boost (MIB), an information-based metric to increase grounding effectiveness without extensive fine-tuning.

**Strengths:**

- Innovative Concept: The approach leverages Multimodal Instance Binding (MIB) to focus on key state grounding, which is a novel method.
- The method has proven to enhance semantic compatibility with instructions, as evidenced by the experiments.

**Weaknesses:**

1. The robotic environment/task as described by the authors does not necessarily need key-state grounding. I believe 100 clean expert demonstrations with end-to-end BC-RNN could achieve similar performances. Key state grounding is useful for long-horizon tasks. I would recommend environments/tasks like [franka Kitchen](https://robotics.farama.org/envs/franka_kitchen/franka_kitchen/) used by BeT and by real-world tasks in [MimicPlay](https://www.notion.so/CoRL2023-reviews-92631182f55f47979d5f15dbdf7c0f7c?pvs=21) . I will change my view if I see more complex robotic applications.

2. If I understand correctly, the LLM in your setup requires task-specific prompts. As described in Appendix A.9.
If the prompt already contains the information needed for the LLM to respond, I don’t see why I should use an LLM. I could directly provide the information to the robot.

**Questions:**

#

1. Is the Exemplar part of the prompt? For example, on page 19:
{Exemplar: For a task that involves picking up a cube and moving it to a goal position…..}
2. Doesn’t the prompt already contain all the information needed for the task? Why do I need to use an LLM? What's the key difference between the exemplar and LLM response?
How many attempts did you make to ask ChatGPT to get the answer you wanted?

---

> ### Author Response · Authors · 2023-11-20
>
> Dear Reviewer S4jr,
>
> Thanks for your questions and constructive suggestions. We also appreciate your acknowledgment of the novelty and effectiveness of the proposed key state grounding algorithm.
>
> We understand that extra baseline comparisons are needed for your re-evaluation of our paper, and some clarifications are needed on the usage of the LLM to alleviate the misunderstanding. Accordingly, we have provided detailed answers to the questions and comments below.
>
> We hope the answers can help finalize your assessment and the rating of our paper. Please also let us know if you have any further questions that we need to provide additional clarifications.
> ***
> **W1.1: The robotic environment/task as described by the authors does not necessarily need key-state grounding. I believe 100 clean expert demonstrations with end-to-end BC-RNN could achieve similar performances.**
>
> **A**: Thanks for the comment. We have performed an additional experiment to provide more information on this point. Specifically, we have experimented with a BC-RNN structure [6] using 500 demonstrations per task for 4 tasks from our environment (listed below).
>
> Both BC-RNN and Decision Transformer (DT) do not employ key states during the training of the policy, while CoTPC and Ours utilize key states to improve the policy training efficiency and generalization.
> We can observe that the BC-RNN structure performs worse than the DT structure when trained using the same 500 demonstrations. Moreover, by employing key states during policy training, the latter two (CoTPC [1] and Ours) outperform BC-RNN and DT by a big margin. Together these comparisons demonstrate that BC-RNN without key state prediction encounters difficulty in efficiently learning the tasks used in our environment. Also, the comparison between CoPTC and Ours demonstrates that the way of using (grounding) key states proposed in our paper is more effective.
>
> Particularly, we would like to clarify that Peg insertion is a complex, multi-stage, long-horizon task, and we have observed that other methods do not achieve high success rates on such long-horizon tasks. Moreover, our method shows about a 30% improvement over the SOTA baseline (CoTPC), both in seen and unseen environments.
> | Algorithm         | P&P Cube | Stack Cube | Turn Faucet | Peg Insertion | Mean |
> |-------------------|----------|------------|-------------|---------------|------|
> | BC-RNN [6]        | 62.7     | 7.8        | 43.2        | 3.2           | 29.2 |
> | Decision Transformer | 65.4  | 13.0       | 39.4        | 5.6           | 30.9 |
> | CoTPC             | 75.2     | 58.8       | 56.4        | 52.8          | 60.8 |
> | Ours              | 89.4     | 83.2       | 67.6        | 75.4          | 78.9 |
>
>
> **W1.2: Key state grounding is useful for long-horizon tasks. I would recommend environments/tasks like Franka Kitchen used by BeT and by real-world tasks in MimicPlay. I will change my view if I see more complex robotic applications.**
>
> **A**: Thanks for your suggestion. We have further experimented with the mentioned environments/tasks to demonstrate that key states aid in policy learning.
>
> For the Franka Kitchen task, where the success rate is measured in completing a specified sequence of multiple subtasks, the Decision Transformers (DT) with and without key states are compared.
>
> We can observe that the DT with key states demonstrates competitive performance in seen environments, showing better learning efficiency of the policy when supplied with the key states. Moreover, we can observe the DT trained with key states performs much better than the DT w/o key states in unseen environments, achieving a relative 73.7% improvement. This indicates that key state guidance has a substantial positive impact on the generalization of the learned policy for complex long-horizon tasks. The results are listed below.
>
> | Franka Kitchen (Success rate) | Seen environment | Unseen environment |
> |-------------------------------|------------------|--------------------|
> | Without key states            | 78.4             | 13.7               |
> | With key states               | 81.8             | 23.8               |
>
>
> We have also examined the MimicPlay dataset and attempted to perform similar experiments. However, we found that MimicPlay focuses on transferring human policy to robot policy with a large number of human demonstrations, which is currently out of our scope. But we would like to thank the reviewer for pointing this out, which can actually be an interesting future research direction for leveraging key state grounding for policy transfer.
> ***

---

> > ### Author Response · Authors · 2023-11-20
> >
> > **W2: If I understand correctly, the LLM in your setup requires task-specific prompts. As described in Appendix A.9. If the prompt already contains the information needed for the LLM to respond, I don't see why I should use an LLM. I could directly provide the information to the robot.**
> >
> > **A**: We would like to clarify that the LLM in our setup does not require task-specific prompts. In other words, throughout the whole experiment, a single in-context learning example is provided for all tasks to bias the LLM toward describing the key state using more physically relevant language. So we do not need to change the example for a different task.
> >
> > In Appendix A.9, the prompt consists of a task description (e.g., what this task is, specified with a single sentence), while the other parts are fixed and provide basic contextual information to the robot. These fixed parts do not require any additional modification for different robot tasks; the only part that needs modification is the task description.
> >
> > Specifying the task description is crucial because, within the same environment, the robot's actions vary depending on different objectives. Recent research [2,3] also validates this point. However, we could improve this part by asking a more powerful foundation model to automatically come up with the tasks to perform in a scene, e.g., GPT-4V, which introduces a visual perception module and could utilize its visual perception capability to formulate an achievable task. This is out of the scope of our current work as we mainly focus on grounding key states related to a task but not proposing tasks in the scene. However, we would like to explore a fully automatic pipeline for proposing tasks by large models in the future.
> > ***
> > **Q1: Is the Exemplar part of the prompt? For example, on page 19: {Exemplar: For a task that involves picking up a cube and moving it to a goal position…..}**
> >
> > **A**: Yes, the (in-context learning) exemplar is part of the prompt. In-context exemplar is beneficial, providing effective context to bias more accurate results from Large Language Models (LLMs) to enhance instruction performance. Recent research has also indicated the necessity of in-context examples for LLMs [3,4,5]. Meanwhile, it's important to note that the (single) exemplar part remains fixed across different task settings, eliminating the need for manual modifications.
> > ***
> > **Q2: Doesn’t the prompt already contain all the information needed for the task? Why do I need to use an LLM? What's the key difference between the exemplar and LLM response? How many attempts did you make to ask ChatGPT to get the answer you wanted?**
> >
> > **A**: It seems there may be a slight misunderstanding. To clarify, the prompt does not contain all the information needed for the task. It uses a single in-context example to enhance the instruction quality, but this single example is fixed across all the tasks we have trained on, thus it does not give all the information needed. The only task-specific thing in the prompt is the task description (i.e., tell the robot what to do), which we must have since the LLM needs to know the task to propose relevant key state descriptions. Moreover, LLM acts in the role of proposing relevant key states for our pipeline to ground.
> >
> > The key difference between the exemplar and the LLM response is that the exemplar serves merely as a fixed template for the output across different tasks, whereas LLM responses can vary depending on the task at hand in proposing task-relevant key states. Our experiments showed that it **only requires one attempt** for GPT to generate the answer we wanted.
> >
> > [1] Jia, Zhiwei, et al. "Chain-of-Thought Predictive Control." arXiv preprint arXiv:2304.00776 (2023).
> >
> > [2] Huang, Wenlong, et al. "Voxposer: Composable 3d value maps for robotic manipulation with language models." arXiv preprint arXiv:2307.05973 (2023).
> >
> > [3] Yu, Wenhao, et al. "Language to Rewards for Robotic Skill Synthesis." arXiv preprint arXiv:2306.08647 (2023).
> >
> > [4] Wei, Jason, et al. "Chain-of-thought prompting elicits reasoning in large language models." Advances in Neural Information Processing Systems 35 (2022): 24824-24837.
> >
> > [5] Yao, Shunyu, et al. "Tree of thoughts: Deliberate problem solving with large language models." arXiv preprint arXiv:2305.10601 (2023).
> >
> > [6] Mandlekar, Ajay, et al. "What matters in learning from offline human demonstrations for robot manipulation." arXiv preprint arXiv:2108.03298 (2021).

---

> > > ### Comment · Reviewer_S4jr · 2023-12-03
> > >
> > > Thank you for the updates and the additional experiments conducted using the Franka Kitchen setup. Your efforts in exploring long-horizon tasks are commendable.
> > > However, I have some concerns regarding the use of Language Models (LLMs) still. You mentioned that the prompt given to the robot includes a task description along with basic contextual information. My understanding is that if this prompt already specifies the "key state" (i.e., the task description), then employing an LLM might be redundant. The strength of LLMs, as you've noted, lies in their ability to autonomously generate tasks to perform in a given scene. But if the task description is already provided, directly inputting this information to the robot could be more straightforward.
> > > Furthermore, as the task description is the only task-specific component in the prompt, it reaffirms my viewpoint. If the purpose of using LLM is to inform the robot of what specific task to execute, and this information is already present in the prompt, the utility of LLM in this scenario becomes questionable.

---

> ### Author Response · Authors · 2023-11-22
>
> Dear Reviewer S4jr,
>
> As time passes by, could you take a look at our rebuttal and let us know whether it resolves your concerns? We are also here to answer any further questions you may have.
>
> Best,
>
> The Authors

---

### Author Response · Authors · 2023-11-20

We sincerely thank all the reviewers for their valuable feedback and constructive comments. It is gratifying to see that the reviewers recognize the novelty of our approach, which offers a principled perspective for key state grounding in various applications (R#1, R#2, R#3, R#4). We are particularly pleased with the acknowledgment of our efforts to enhance semantic compatibility in foundation models and to improve grounding efficiency (R#1, R#2, R#4). The positive feedback on our method's ability to improve generalization in robotic tasks (R#2) is highly encouraging. Additionally, we appreciate the recognition that our method is clear and straightforward to follow (R#4), which is essential for its widespread adoption and replication in future research.

We have meticulously addressed each specific and valuable point raised by the reviewers, providing detailed explanations and supplementary information, as well as additional experimental demonstrations where necessary. The manuscript has been updated considering the reviewers' comments, with revisions highlighted in blue.

We have responded to each reviewer's queries and concerns, aiming to clarify our approach and contribution. Our sincere thanks to all reviewers for their valuable feedback, which has significantly improved our work. We are open to any further questions or clarifications needed to enhance our work. Please do not hesitate to contact us for any additional information or explanations.

Thank you for your review!

---

### Author Response · Authors · 2023-11-23

Dear Reviewers and ACs,

We would like to clarify that our work is not to solve the learning of manipulation tasks. But to propose a way to help reduce the annotation effort needed for grounded key states.

Specifically, since manipulation concepts are important for efficient learning and generalization of interaction policies, the contribution of our work mainly lies in using LLMs to automatically propose more fine-grained ones (compared to human-labeled ones) that can help with policy learning, while greatly minimizing the annotation effort (of grounded states).

Our experiments verify that the proposed MIB criterion is effective on various tasks from ManiSkill2 and Franka Kitchen. We agree that fully relying on LLMs is not healthy, which is something we are not proposing here. Again, we just want to show that there is an alternative to human labeling of those grounded states, and we have verified its effectiveness.

---

### Meta-Review · Area_Chair_BMnk · 2023-12-18

**Metareview:**

The authors introduce an approach to identifying key states in a robot trajectory.  The intention is to leverage LLMs to produce potential states and paraphrases which can be aligned to "bottlenecks" in the trajectory that are semantically useful to task completion. Given the generally weak nature of the scores, I also read the paper. I must admit that I agree that the paper was difficult to follow, hard to decipher which are the key insights provided here, and where/why the approach works and by how much.  There are lots of comparisons (both in the paper and in the review responses) to the same approaches and with very different margins and it's difficult to decipher where/why the approach is working best -- particularly in relation to DT and CoTPC.

**Justification For Why Not Higher Score:**

I quite like the intuition of the work as I understand it, but think that it requires integration of everything from the review process, a rewrite, and clearer exposition around why key decisions (e.g. w/ vs w/out key frames, cases where perception-free LLMs should fail, ...).  That is too many requested changes to accept in the current form.

**Justification For Why Not Lower Score:**

N/A

---

### Decision · Program_Chairs · 2024-01-16

Reject